# Maternal circulating miRNAs that predict infant FASD outcomes influence placental maturation

Alexander M Tseng[1], Amanda H Mahnke[1], Alan B Wells[2,3], Nihal A Salem[1], Andrea M Allan[4], Victoria HJ Roberts[5], Natali Newman[6], Nicole AR Walter[6], Christopher D Kroenke[6], Kathleen A Grant[6], Lisa K Akison[7], Karen M Moritz[7], Christina D Chambers[2,3], Rajesh C Miranda[1], Collaborative Initiative on Fetal Alcohol Spectrum Disorders

Prenatal alcohol exposure (PAE), like other pregnancy complications, can result in placental insufficiency and fetal growth restriction, although the linking causal mechanisms are unclear. We previously identified 11 gestationally elevated maternal circulating miRNAs ($_{HEa}$miRNAs) that predicted infant growth deficits following PAE. Here, we investigated whether these $_{HEa}$miRNAs contribute to the pathology of PAE, by inhibiting trophoblast epithelial–mesenchymal transition (EMT), a pathway critical for placental development. We now report for the first time that PAE inhibits expression of placental pro-EMT pathway members in both rodents and primates, and that $_{HEa}$miRNAs collectively, but not individually, mediate placental EMT inhibition. $_{HEa}$miRNAs collectively, but not individually, also inhibited cell proliferation and the EMT pathway in cultured trophoblasts, while inducing cell stress, and following trophoblast syncytialization, aberrant endocrine maturation. Moreover, a single intravascular administration of the pooled murine-expressed $_{HEa}$miRNAs, to pregnant mice, decreased placental and fetal growth and inhibited the expression of pro-EMT transcripts in the placenta. Our data suggest that $_{HEa}$miRNAs collectively interfere with placental development, contributing to the pathology of PAE, and perhaps also, to other causes of fetal growth restriction.

## Introduction

Prenatal alcohol exposure (PAE) is common (1, 2, 3). Between 1.1% and 5% of school children in the United States are conservatively estimated to have a fetal alcohol spectrum disorder (FASD) (4). Consequently, FASD, due to PAE, is the single largest cause of developmental disabilities in the United States and worldwide (5) and a comorbid factor in a number of other prevalent developmental neurobehavioral disabilities, including attention deficit/hyperactivity and autism spectrum disorders (6).

PAE can result in decreased body weight, height, and/or head circumference in infants. Consequently, infant growth deficits are a cardinal diagnostic feature for fetal alcohol syndrome (7), which represents the severe end of the FASD continuum. However, although well recognized as a diagnostic feature, the mechanistic linkage between PAE and growth restriction remains unclear. In 2016, as part of our effort to identify maternal diagnostic biomarkers of the effect of PAE, we reported that elevated levels of 11 distinct miRNAs in maternal circulation during the second and third trimesters distinguished infants who were affected by in utero alcohol exposure (heavily exposed affected [HEa]) from those who were apparently unaffected at birth by PAE (heavily exposed unaffected [HEua]) or those who were unexposed (UE) (8). In that study, we predicted, based on bioinformatics analyses, that these $_{HEa}$miRNAs (MIMAT0004569 [hsa-miR-222-5p], MIMAT0004561 [hsa-miR-187-5p], MIMAT0000687 [hsa-miR-299-3p], MIMAT0004765 [hsa-miR-491-3p], MIMAT0004948 [hsa-miR-885-3p], MIMAT0002842 [hsa-miR-518f-3p], MIMAT0004957 [hsa-miR-760], MIMAT0003880 [hsa-miR-671-5p], MIMAT0001541 [hsa-miR-449a], MIMAT0000265 [hsa-miR-204-5p], and MIMAT0002869 [hsa-miR-519a-3p]) could influence signaling pathways crucial for early development, particularly the epithelial–mesenchymal transition (EMT) pathway.

Placental development involves maturation of cytotrophoblasts at the tips of anchoring villi into invasive extravillous trophoblasts, as well as fusion of cytotrophoblasts into multinucleate, hormone-producing syncytiotrophoblasts (9). Maturation into extravillous trophoblasts, which invade the maternal decidua and remodel the uterine spiral arteries into low-resistance high-flow vessels that enable optimal perfusion for nutrient and waste exchange, requires cytotrophoblasts to undergo EMT (10). Impaired placental EMT, as

[1]Department of Neuroscience and Experimental Therapeutics, Texas A&M University Health Science Center, Bryan, TX, USA   [2]Clinical and Translational Research Institute, University of California San Diego, San Diego, CA, USA   [3]Department of Pediatrics, University of California San Diego, San Diego, CA, USA   [4]Department of Neurosciences, University of New Mexico, Albuquerque, NM, USA   [5]Division of Reproductive and Developmental Sciences, Oregon National Primate Research Center, Oregon Health & Science University, Portland, OR, USA   [6]Division of Neuroscience, Oregon National Primate Research Center, Oregon Health & Science University, Portland, OR, USA   [7]Child Health Research Centre and School of Biomedical Sciences, The University of Queensland, Brisbane, Australia

Correspondence: miranda@medicine.tamhsc.edu; chchambers@ucsd.edu
Rajesh C Miranda and Christina D Chambers are co-senior authors.

well as orchestration of the opposing mesenchymal–epithelial transition pathway, has been found in conditions resulting from placental malfunction, primarily preeclampsia (11, 12, 13, 14, 15, 16). Although there have been no previous studies directly investigating the effects of PAE on placental EMT, a rodent study demonstrated that PAE, during a broad developmental window, reduced the number of invasive trophoblasts within the mesometrial triangle, a region of the uterine horn directly underlying the decidua (17). Furthermore, both human and rodent studies have found PAE disrupts placental morphology and interferes with cytotrophoblast maturation, as with preeclampsia (18, 19, 20, 21). Disrupted trophoblast maturation, seen in these conditions, is associated with aberrant expression of placental hormones, primarily human chorionic gonadotropin (hCG) (22, 23, 24, 25).

Our study is the first to report that PAE interferes with expression of core placental EMT pathway members. Using rodent and primate models of gestation, as well as complementary miRNA overexpression and knockdown studies in vitro, we also provide evidence that $_{HEa}$miRNAs, which predict infant growth deficits due to PAE, collectively but not individually, mediate PAE's effects on placental EMT through their effects on cytotrophoblast maturation and cellular stress. In a mouse model of pregnancy, a single combined exposure to the murine-expressed $_{HEa}$miRNAs resulted in placental EMT inhibition and diminished placental and fetal growth. Collectively, these data suggest that elevated $_{HEa}$miRNAs may represent an emergent maternal stress response that triggers fetal growth restriction, although subgroups of $_{HEa}$miRNAs may compete to protect against the loss of EMT. Moreover, most members of the group of $_{HEa}$miRNAs have also been implicated in other placental insufficiency and growth restriction syndromes, giving rise to the possibility that growth restriction syndromes may share common etiological mediators.

# Results

## $_{HEa}$miRNAs are implicated in placental-associated pathologies

Given our prediction that $_{HEa}$miRNAs interfere with signaling pathways governing fetal and placental development (8), we conducted a literature review of reports on $_{HEa}$miRNA levels in gestational pathologies caused by poor placentation (26, 27, 28). Surprisingly, placental and plasma levels of 8 of 11 $_{HEa}$miRNAs were significantly dysregulated in one or more of these gestational pathologies with expression of the majority of these eight miRNAs altered in both fetal growth restriction and preeclampsia (Fig 1A) (29, 30, 31, 32, 33, 34, 35, 36, 37, 38, 39, 40, 41, 42, 43, 44, 45, 46, 47, 48, 49), both of which are characterized by poor placental invasion (50, 51, 52, 53, 54, 55, 56).

## $_{HEa}$miRNAs explain variance in infant growth outcomes due to PAE

Given the association of individual $_{HEa}$miRNAs with gestational pathologies, we sought to determine if circulating $_{HEa}$miRNA levels could explain the variance in sex and gestational age–adjusted neonatal height, weight, and head circumference in our Ukrainian birth cohort, which are growth measures sensitive to in utero

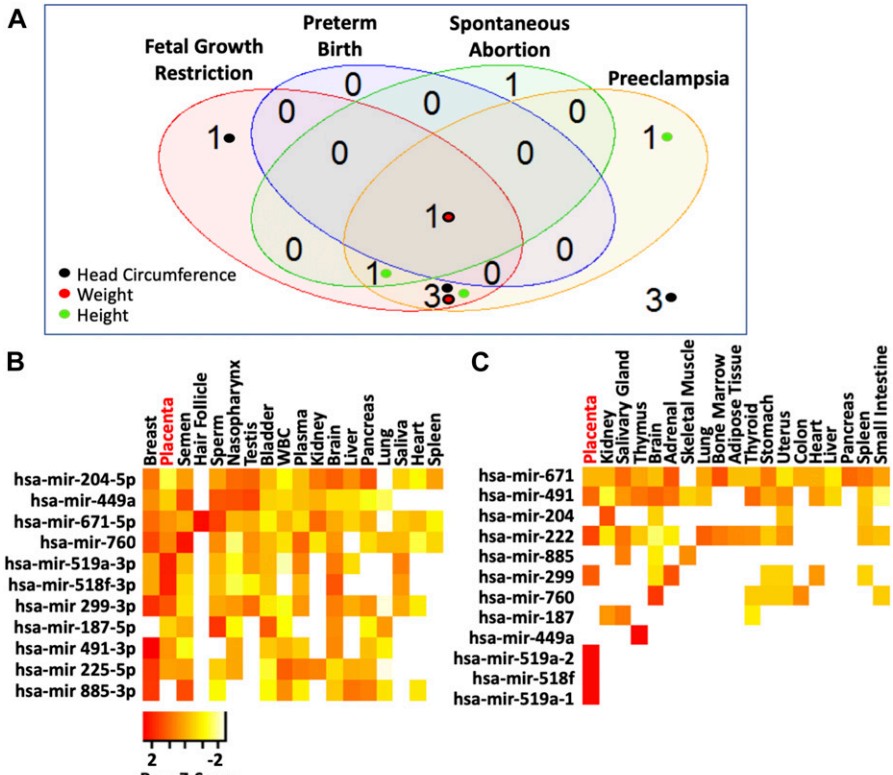

**Figure 1.** $_{HEa}$**miRNAs are placentally enriched and associated with gestational pathologies.**
**(A)** Venn diagram on number of $_{HEa}$miRNAs reported to be associated with different gestational pathologies. Inset colored circles represent the corresponding sex and gestational age–adjusted growth parameters these miRNAs were correlated with. Of the 22 studies queried, 11 (50%) used unbiased screenings for miRNA expression. **(B, C)** Heat map of mature $_{HEa}$miRNA expression (B) and pri-$_{HEa}$miRNA expression (C) across different tissues resulting from secondary analysis of publicly available RNA-sequencing data. Legend depicts row-centered Z-score.

**Table 1.** HEamiRNAs are significantly correlated with independent measures of infant size.

| MIMAT no. | miRNA | Trimester | Weight | | | Height | | | Head circumference | | |
|---|---|---|---|---|---|---|---|---|---|---|---|
| | | | Sig. | $R^2$ | $\rho$ | Sig. | $R^2$ | $\rho$ | Sig. | $R^2$ | $\rho$ |
| MIMAT0004569 | hsa-miR-222-5p | 2 | 0.821 | 1.224 | −0.051 | 0.066 | 9.572 | −0.179 | 0.8 | 1.732 | −0.104 |
| MIMAT0004561 | hsa-miR-187-5p | 2 | 0.462 | 6.347 | 0.068 | 0.17 | 12.607 | −0.074 | 0.134 | 10.903 | 0.103 |
| **MIMAT0000687** | **hsa-miR-299-3p** | 2 | 0.552 | 1.113 | 0.029 | 0.069 | 9.299 | −0.203 | **0.036**[a] | **8.65** | **0.1** |
| **MIMAT0004765** | **hsa-miR-491-3p** | 2 | 0.172 | 3.61 | 0.112 | 0.849 | 2.033 | −0.055 | **0.024**[a] | **12.529** | **0.156** |
| **MIMAT0004948** | **hsa-miR-885-3p** | 2 | 0.142 | 4.227 | −0.174 | **0.044**[a] | **7.667** | **−0.231** | 0.59 | 1.36 | −0.115 |
| **MIMAT0002842** | **hsa-miR-518f-3p** | 2 | 0.246 | 2.517 | 0.134 | 0.918 | 2.134 | −0.118 | **0.007**[b] | **14.561** | **0.219** |
| MIMAT0004957 | hsa-miR-760 | 2 | 0.059 | 6.314 | 0.195 | 0.22 | 4.096 | 0.079 | 0.055 | 10.158 | 0.195 |
| MIMAT0003880 | hsa-miR-671-5p | 2 | 0.123 | 7.24 | 0.11 | 0.578 | 5.264 | −0.031 | 0.073 | 10.794 | 0.107 |
| MIMAT0001541 | hsa-miR-449a | 2 | 0.101 | 11.584 | 0.104 | 0.718 | 5.851 | −0.072 | 0.173 | 10.036 | 0.068 |
| **MIMAT0000265** | **hsa-miR-204-5p** | 2 | **0.026**[a] | **12.377** | **0.184** | 0.272 | 4.973 | 0 | 0.131 | 7.095 | 0.108 |
| **MIMAT0002869** | **hsa-miR-519a-3p** | 2 | **0.034**[a] | **7.975** | **0.153** | 0.403 | 6.83 | −0.012 | 0.093 | 8.181 | 0.096 |
| **MIMAT0004569** | **hsa-miR-222-5p** | 3 | 0.875 | 0.993 | −0.046 | **0.018**[a] | **10.709** | **−0.196** | 0.577 | 4.696 | −0.01 |
| MIMAT0004561 | hsa-miR-187-5p | 3 | 0.538 | 2.055 | 0.049 | 0.37 | 2.029 | −0.109 | 0.784 | 3.697 | 0.002 |
| MIMAT0000687 | hsa-miR-299-3p | 3 | 0.511 | 0.762 | 0.005 | 0.514 | 1.769 | −0.072 | 0.87 | 3.786 | −0.077 |
| MIMAT0004765 | hsa-miR-491-3p | 3 | 0.824 | 3.165 | −0.028 | 0.2 | 12.122 | −0.121 | 0.747 | 4.188 | −0.081 |
| MIMAT0004948 | hsa-miR-885-3p | 3 | 0.807 | 0.148 | 0.029 | 0.102 | 4.686 | −0.156 | 0.376 | 5.009 | 0.032 |
| MIMAT0002842 | hsa-miR-518f-3p | 3 | 0.515 | 2.099 | 0.109 | 0.421 | 1.715 | 0.016 | 0.245 | 7.917 | 0.152 |
| MIMAT0004957 | hsa-miR-760 | 3 | 0.368 | 1.396 | 0.141 | 0.761 | 0.716 | −0.022 | 0.207 | 6.052 | 0.172 |
| MIMAT0003880 | hsa-miR-671-5p | 3 | 0.055 | 8.715 | 0.155 | 0.367 | 3.521 | −0.133 | 0.076 | 8.196 | 0.15 |
| **MIMAT0001541** | **hsa-miR-449a** | 3 | 0.995 | 0.085 | −0.06 | 0.982 | 0.678 | −0.151 | **0.026**[a] | **12.022** | **0.135** |
| **MIMAT0000265** | **hsa-miR-204-5p** | 3 | **0.019**[a] | **11.872** | **0.23** | 0.206 | 5.589 | 0.022 | **0.002**[b] | **18.683** | **0.319** |
| MIMAT0002869 | hsa-miR-519a-3p | 3 | 0.391 | 2.82 | 0.043 | 0.302 | 5.917 | −0.151 | 0.106 | 9.286 | 0.118 |

The correlation of the second and third trimester maternal plasma HEamiRNA levels with independent measures of infant size. HEamiRNAs and their significantly correlated sex and gestational age–adjusted growth parameters appear in bold. $R^2$ is expressed as the percentage (×100) of variance explained.
[a]$P < 0.05$.
[b]$P < 0.01$.

environment (57). We found that eight of the HEamiRNAs each significantly explained between 7% and 19% of infant variation in these growth measures (Table 1). Furthermore, seven of these miRNAs were also associated with fetal growth restriction and preeclampsia as identified by our literature review (Fig 1A). Interestingly, a multivariate statistical regression model that accounted for levels of all 11 HEamiRNAs together, explained a far greater proportion of infant variance, between 24% and 31%, in all three growth measures than accounting for them individually (Table S1), suggesting HEamiRNAs collectively account for the variance in infant growth outcomes.

### HEamiRNAs are transcribed preferentially in the placenta

Data extracted from publicly available gene expression profiling datasets (58) show that HEamiRNAs and their unprocessed precursor transcripts, HEapri-miRNAs, are enriched in placenta compared with other tissues, suggesting that the placenta itself transcribes these miRNAs and may be a significant contributory tissue to maternal circulating HEamiRNAs (Fig 1B and C). Moreover,

because HEamiRNAs are also associated with gestational pathologies caused by poor placental invasion, these HEamiRNAs may also contribute to the placental response to PAE. We, therefore, assessed in rodent and primate models, whether PAE could result in impaired EMT, and if HEamiRNAs could explain the effects of PAE on placental EMT-associated gene expression.

### HEamiRNAs moderate placental EMT impairment in PAE models

EMT, in trophoblasts, is characterized by the disappearance of epithelial markers such as E-Cadherin and the appearance of mesenchymal markers such as the intermediate filament, vimentin, a process that is controlled by the expression of key mesenchymal determination transcription factors, Snail1 and 2 and TWIST, as extensively described (10, 14, 15, 59, 60, 61, 62). These five markers have been used to assess EMT in a variety of model systems, so our studies used these markers to assess the effects of alcohol and HEamiRNAs on trophoblast EMT.

In the first analysis, using a murine model of PAE that mimicked moderate to binge-type alcohol consumption throughout early and

mid-pregnancy, we fractionated GD14 placenta into three zones: the cytotrophoblast- and syncytiotrophoblast-rich labyrinth zone, the glycogen- and spongiotrophoblast-rich junctional zone, and the

decidual zone comprising the endometrial contribution to the placenta (Fig 2A). Multivariate analysis of variance (MANOVA) for expression of these five core genes in the EMT pathway within

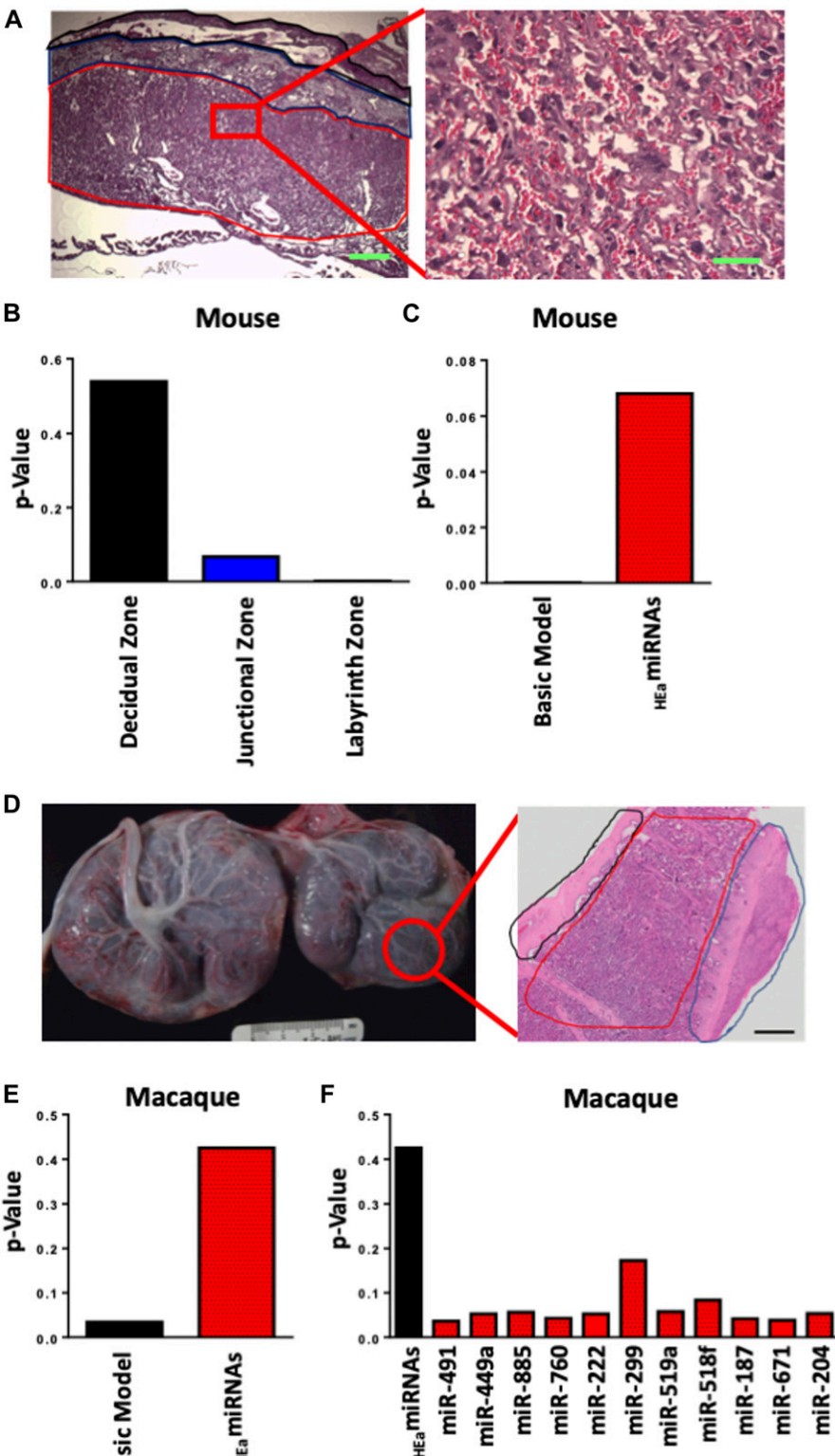

**Figure 2.** $_{HEa}$miRNAs mediate the effect of PAE on EMT pathway members in mouse and macaque placentas. **(A)** Histological image of GD14 mouse placenta. Outlined in red is the labyrinth zone, blue is the junctional zone, and black is the decidual zone with the scale bar (green) demarcating 200 $\mu$m. Inset is a high-magnification image of the labyrinth zone with the scale bar (green) demarcating 50 $\mu$m. **(B)** MANOVA of gene expression of core EMT pathway members in different regions of the mouse placenta in control and PAE mice (n = 29 samples). **(C)** MANCOVA of gene expression of core EMT pathway members in the mouse placental labyrinth zone before (Basic Model) and after accounting for the expression of $_{HEa}$miRNAs (n = 29 samples). **(D)** Gross anatomy photograph of the primary (left) and secondary (right) lobes of a GD135 macaque placenta. Outlined in red is an individual cotyledon from the secondary lobe. Inset is a full thickness hematoxylin and eosin–stained histological section of a representative cotyledon with the fetal membranes outlined in black, villous tissue outlined in red. and maternal decidua in blue. Ruler is 3 cm and scale bar (black) is 2 mm. **(E)** MANCOVA of gene expression of core EMT pathway members in placental cotyledons of PAE and control macaques, accounting for the expression of $_{HEa}$miRNAs collectively (n = 23 samples). **(F)** MANCOVA of gene expression of core EMT pathway members in macaque placentas after accounting for expression of $_{HEa}$miRNAs individually (n = 23 samples).

placental trophoblasts revealed a significant effect of ethanol exposure on EMT pathway member expression selectively within the labyrinth zone (Pillai's trace statistic, $F_{(5,21)}$ = 6.85, $P$ < 0.001, Fig 2B) but not within the junctional or decidual zones. Post hoc univariate ANOVA indicated ethanol exposure specifically elevated *CDH1* ($F_{(1,25)}$ = 7.452, $P$ = 0.011), which encodes epithelial E-Cadherin, whereas expression of the pro-mesenchymal transcription factor *SNAI1*, which encodes Snail1, was significantly reduced ($F_{(1,25)}$ = 21.022, $P$ = 0.0001). We also observed a significant interaction between fetal sex and PAE on expression of *SNAI2*, which encodes Snail2 ($F_{(1,25)}$ = 2.18, $P$ = 0.047) and a trend towards decreased expression of the terminal

mesenchymal marker *VIM* (vimentin, $F_{(1,25)}$ = 2.749, $P$ = 0.11), whereas there was no effect on *TWIST* expression (Fig 3A–E). Consistent with our gene expression data, E-Cadherin protein levels were significantly elevated in the labyrinth zone of PAE placenta ($F_{(1,24)}$ = 31.63, $P$ = 0.0005), whereas not in the junctional or decidual zones (Figs 3F and S1A and B). However, when we controlled for expression of the eight mouse homologs of HEamiRNAs as a covariate, using multivariate analysis of covariance (MANCOVA), ethanol's effect on EMT became marginally nonsignificant (Pillai's trace, $F_{(5,21)}$ = 2.713, $P$ = 0.068) (Fig 2C), suggesting that these miRNAs partially mediate effects of PAE on EMT pathway members in mice. Interestingly, PAE

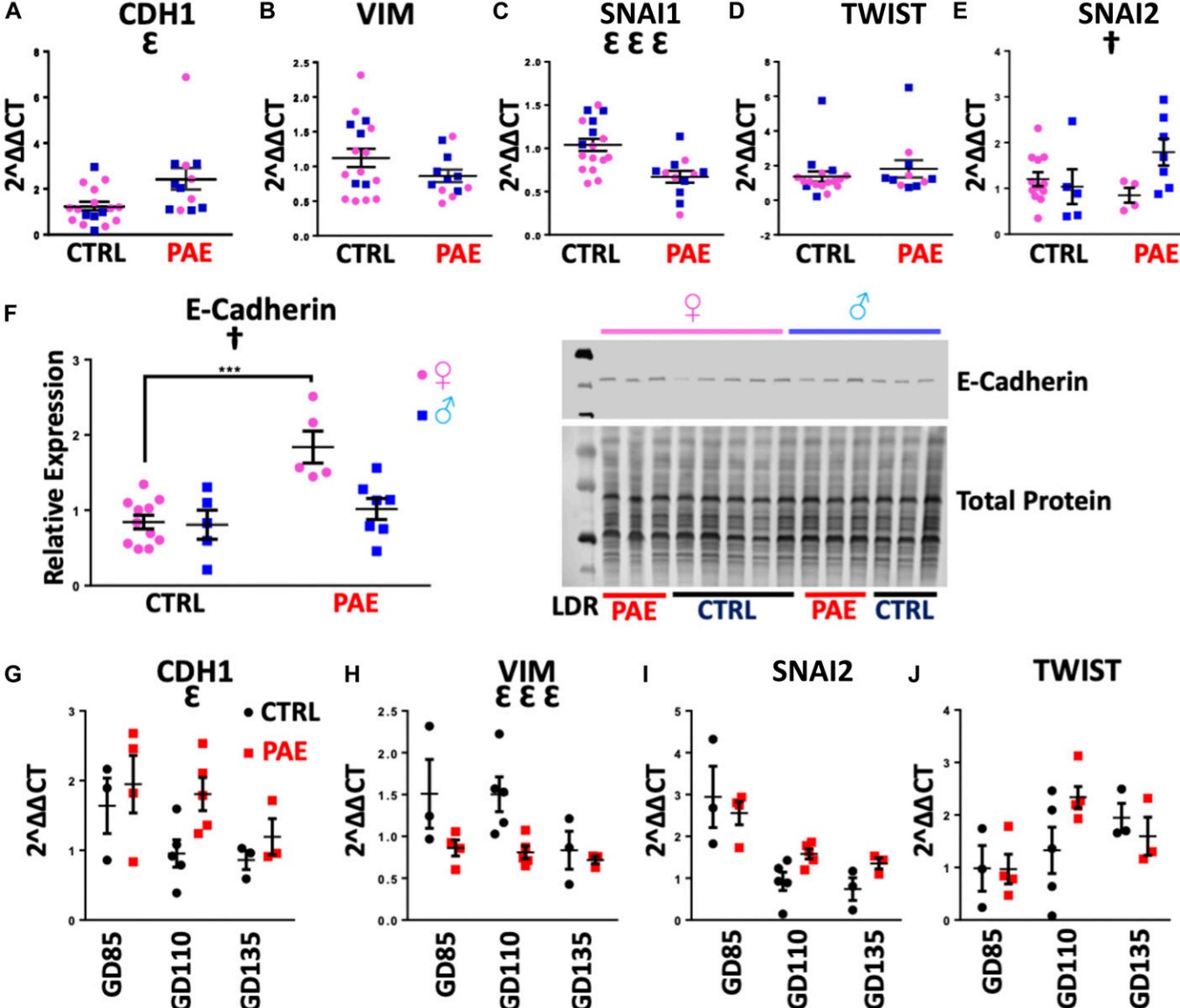

**Figure 3. PAE interferes with the EMT pathway in mouse and macaque placentas.**
**(A–E)** Expression of *CDH1* (A), *VIM* (B), *SNAI1* (C), *TWIST* (D), and *SNAI2* (E) in the placental labyrinth zone of PAE and control mice (n = 5–12 samples per group). **(F)** Densitometric quantification of E-Cadherin expression in the labyrinth zone of PAE and control mice as well as representative blot of E-Cadherin expression and total protein expression (right, n = 5–12 samples per group). **(G–J)** Expression of *CDH1* (G), *VIM* (H), *SNAI2* (I), and *TWIST* (J) transcripts in PAE and control macaque placental cotyledons (n = 3–5 samples per group). Results are expressed as the mean ± SEM, LDR = molecular weight ladder; ANOVA: significant main effect of PAE ($^{\varepsilon}P$ < 0.05, $^{\varepsilon\varepsilon\varepsilon}P$ < 0.001), significant interaction effect (sex by PAE, [$^{\dagger}P$ < 0.05]). For post hoc analysis, $^{***}P$ < 0.001 by Tukey's HSD.

limited to the periconceptional period in rats also influenced expression of EMT core transcripts (Figs S2B and S3A–E).

To determine if PAE's effects on EMT pathway members in placenta are broadly conserved throughout mammalian evolution, we adopted a nonhuman primate (macaque) model of moderate to binge-type alcohol consumption. Placental tissues were isolated from GD85, GD110, and GD135 placenta (Fig 2D), which spans the human equivalent of the mid-second to mid-third trimester (Fig S2C). There was a significant effect of ethanol exposure on expression of core EMT mRNA transcripts by MANOVA (Pillai's trace statistic, $F_{(4,9)}$ = 4.229 $P$ = 0.045, Fig 3B). Consistent with our findings in mouse, post hoc univariate ANOVA indicated that in primate placenta, ethanol exposure significantly increased $CDH1$ expression ($F_{(1,12)}$ = 4.866, $P$ = 0.048), whereas $VIM$ expression was significantly reduced ($F_{(1,12)}$ = 12.782, $P$ = 0.0004), suggesting that, as in the mouse, PAE also impairs EMT in the primate placenta. Interestingly, there was no effect on $SNAI2$ or $TWIST$ expression (Fig 3G–J). As in mice, accounting for expression of $_{HEa}$miRNAs together as a covariate abolished the significant effect of PAE on EMT, although to a greater degree than mice (Pillai's trace, $F_{(1,1)}$ = 1.605, $P$ = 0.425, Fig 2E). Interestingly, accounting for expression of individual $_{HEa}$miRNAs did not explain the effects of PAE on placental EMT, suggesting that $_{HEa}$miRNAs act in concert to mediate the effect of PAE on EMT in the primate placenta (Fig 2F).

Collectively, our data suggest PAE-induced impairment of EMT in the trophoblastic compartment of placentae is conserved between rodents and nonhuman primates and that $_{HEa}$miRNAs, particularly in primates, may moderate the effect of PAE on placental EMT. Consequently, subsequent studies focused on the collective role of $_{HEa}$miRNAs, either on basal or on alcohol-influenced placental trophoblast growth, invasion, and the maturation of physiological function.

### $_{HEa}$miRNAs impair EMT in a model of human cytotrophoblasts

To investigate whether $_{HEa}$miRNAs collectively interfere with the EMT pathway, as suggested by our in vivo data, we examined the effects of transfecting $_{HEa}$miRNA mimics and antagomirs into BeWO cytotrophoblasts (Fig 4A). We initially overexpressed each of the 11 $_{HEa}$miRNAs individually, to determine whether any of them could influence the EMT pathway. We did not observe any significant effects (Fig S4), consistent with our findings in the primate PAE model that individual miRNAs did not explain the effects of ethanol on EMT. In contrast, transfection of pooled $_{HEa}$miRNAs into cytotrophoblasts significantly increased $CDH1$ expression ($F_{(1,36)}$ = 30.08, $P$ < 0.0001). Interestingly, expression of the pro-mesenchymal transcription factors $TWIST$ and $SNAI1$ were also significantly reduced, but only in the context of concomitant 320 mg/dl ethanol treatment, pointing to an interaction effect between $_{HEa}$miRNAs and ethanol ($F_{(1,36)}$ = 5.650 and 5.146, respectively, $P$ = 0.023 and $P$ = 0.029, Fig 4B–E). Consistent with our qPCR data, transfection of $_{HEa}$miRNAs also significantly increased E-Cadherin protein expression ($F_{(1,20)}$ = 33.86, $P$ < 0.0001, Fig 4F). We were unable to detect $SNAI2$ transcript expression or vimentin protein expression in these cells, consistent with previous reports (63).

We next sought to determine if more restricted subsets of $_{HEa}$miRNAs could recapitulate the effects of $_{HEa}$miRNAs collectively on EMT. Thus, we overexpressed hsa-miR-222-5p and hsa-miR-519a-3p,

which are implicated in preeclampsia and fetal growth restriction, as well as hsa-miR-885-3p, hsa-miR-518f-3p, and hsa-miR-204-5p, which are implicated in preeclampsia, fetal growth restriction, and spontaneous abortion or preterm labor (Fig S5A). In contrast to the collective action for all $_{HEa}$miRNAs, exposure to each of these pools resulted in significant decreases in $CDH1$ expression ($F_{(2,12)}$ = 20.12, $P$ = 0.0001). The pool including hsa-miR-885-3p, hsa-miR-518f-3p, and hsa-miR-204-5p also significantly increased Snai1 $SNAI1$ ($F_{(2,12)}$ = 4.604, $P$ = 0.0328; Dunnett's post hoc $P$ = 0.0497, Fig S5B–E). These data suggest that $_{HEa}$miRNAs include subgroups of miRNAs that have the potential to partly mitigate the effects of elevating the entire pool. However, the potential protective effects of these subgroups are masked by the collective function of the entire group of $_{HEa}$miRNAs.

Whereas transfection of $_{HEa}$miRNA mimics increased $CDH1$ expression, transfection of pooled antagomirs to $_{HEa}$miRNAs significantly reduced $CDH1$ expression, only in the context of 320 mg/dl ethanol co-exposure ($_{HEa}$miRNA × 320 mg/dl EtOH interaction, $F_{(1,36)}$ = 13.51, $P$ = 0.0008; post hoc Tukey's honest significance difference (HSD), $P$ = 0.005, Fig 4G). However, expression of $TWIST$ was also decreased with ethanol co-exposure, and there was no significant difference in E-Cadherin protein expression relative to the control (Fig 4H–K). Thus, our data suggest that increasing $_{HEa}$miRNA levels impairs EMT pathway members in cytotrophoblasts, whereas inhibiting their action has a more restricted effect on EMT pathway members.

### $_{HEa}$miRNAs impair EMT in a model of human extravillous trophoblasts

We next investigated the effect of $_{HEa}$miRNAs on EMT in HTR-8/SVneo extravillous trophoblast-type cells (Fig 5A). Transfecting pooled $_{HEa}$miRNA mimics into extravillous trophoblasts significantly decreased $VIM$ expression ($F_{(1,36)}$ = 28.43, $P$ < 0.0001). Expression of pro-mesenchymal transcription factor $SNAI2$ was also reduced ($F_{(1,36)}$ = 64.88, $P$ < 0.0001). As with cytotrophoblasts, expression of $SNAI1$ and $TWIST$ were reduced only with 320 mg/dl ethanol co-exposure ($_{HEa}$miRNA × 320 mg/dl EtOH interaction, $F_{(1,36)}$ = 4.21 and 5.18, $P$ = 0.048 and 0.029, respectively; post hoc Tukey's HSD, $P$ = 0.027 and $P$ < 0.0001, respectively, Fig 5B–E). Consistent with our qPCR data, vimentin protein expression was also significantly reduced ($F_{(1,20)}$ = 9.535, $P$ = 0.006, Fig 5F). Interestingly, there was also a main effect of alcohol exposure on decreasing vimentin protein expression ($F_{(1,20)}$ = 7.303, $P$ = 0.014). We were unable to detect expression of $CDH1$ transcript, or its E-Cadherin protein product, in extravillous trophoblasts, consistent with previous reports (63).

In contrast to $_{HEa}$miRNA mimics, transfecting pooled antagomirs significantly increased $VIM$ expression ($F_{(1,35)}$ = 42.56, $P$ < 0.0001). Likewise, antagomir transfection increased expression of $SNAI2$ in the context of 320 mg/dl ethanol co-exposure and $SNAI1$ under basal conditions ($_{HEa}$miRNA × 320 mg/dl Etoh interaction, $F_{(1,35)}$ = 10.31 and 4.86, $P$ = 0.01 and $P$ = 0.034, respectively; post hoc Tukey's HSD, $P$ < 0.0001, Fig 5G–J). Despite our qPCR data, we did not observe significant differences in vimentin protein expression between treatment groups (Fig 5K). Collectively, our data indicate that increased trophoblastic $_{HEa}$miRNA levels favor an epithelial phenotype, whereas inhibiting their action promotes a mesenchymal phenotype.

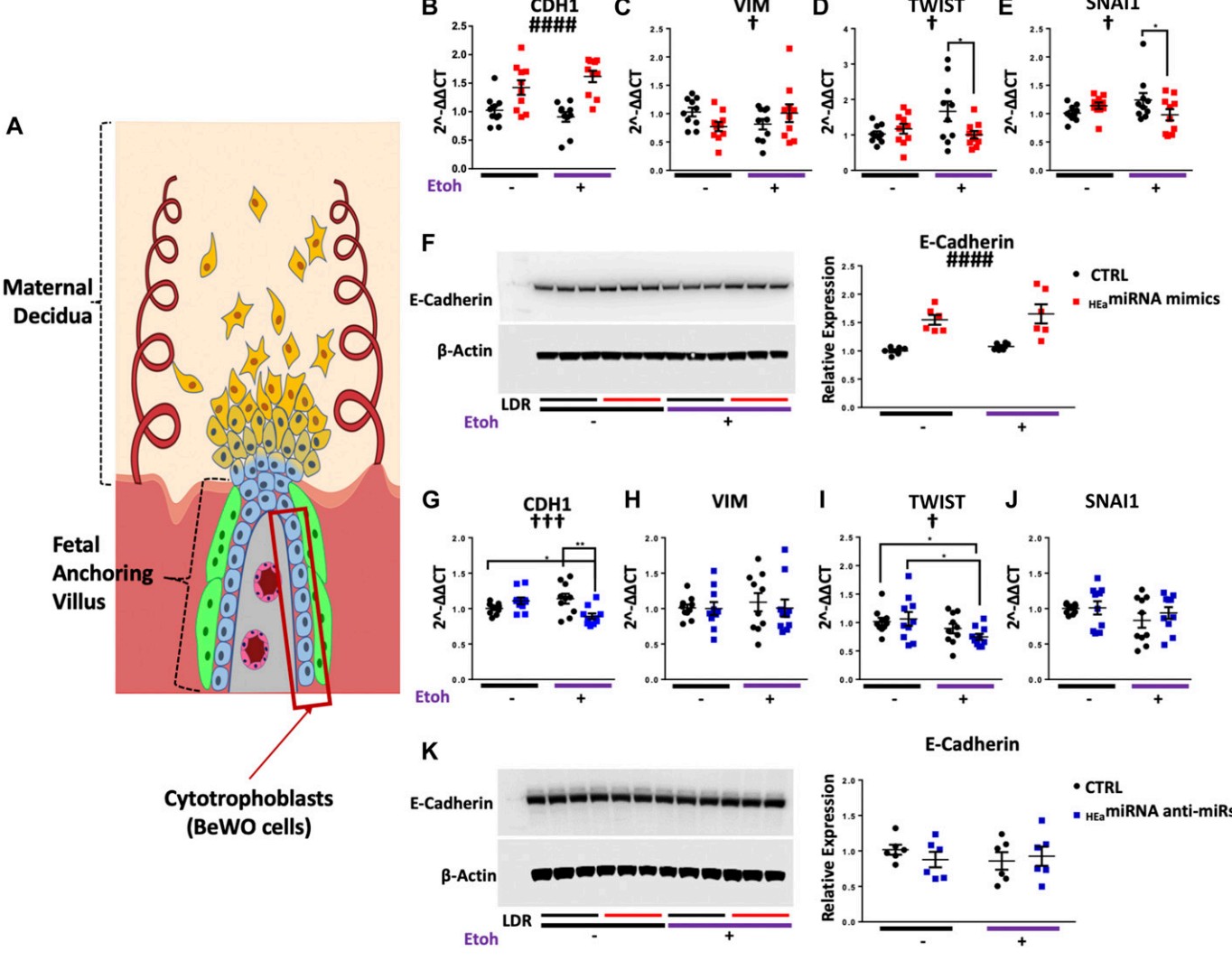

**Figure 4.** ₍HEa₎miRNAs interfere with the EMT pathway in BeWO cytotrophoblasts.
**(A)** Diagram of a placental anchoring villous and maternal decidua with the boxed area denoting cytotrophoblasts. **(B–F)** Expression of *CDH1* (B), *VIM* (C), *TWIST* (D), and *SNAI1* (E) transcripts and densitometric quantification of E-Cadherin protein levels (F) in BeWO cytotrophoblasts following ₍HEa₎miRNA or control miRNA overexpression with or without concomitant 320 mg/dl ethanol exposure. **(G–K)** Expression of *CDH1* (G), *VIM* (H), *TWIST* (I), and *SNAI1* transcripts (J) and densitometric quantification of E-Cadherin protein levels (K) in BeWO cytotrophoblasts following ₍HEa₎miRNAs or control hairpin inhibitor transfection with or without concomitant 320 mg/dl ethanol exposure. Results are expressed as the mean ± SEM, LDR = molecular weight ladder, n = 10 samples per group; ANOVA: significant main effect of ₍HEa₎miRNA transfection ($^{####}P < 0.0001$), significant interaction effect (₍HEa₎miRNA by 320 mg/dl ethanol, [$^{†}P < 0.05$, $^{†††}P < 0.001$]). For post hoc analysis $*P < 0.05$, $**P < 0.01$ by Tukey's HSD.

## Antagomirs prevent ₍HEa₎miRNAs' inhibition of EMT

We next investigated if pretreating cytrophoblasts with pooled ₍HEa₎miRNA antagomirs could prevent inhibition of the EMT pathway caused by transfecting ₍HEa₎miRNA mimics. Pretreatment of cytotrophoblasts with ₍HEa₎miRNA antagomirs prevented the elevation in *CDH1* caused by transfection with ₍HEa₎miRNA mimics (post hoc Tukey's HSD, n = 10 samples per group, $P = 0.004$). Likewise, pre-transfection with ₍HEa₎miRNA antagomirs also prevented ₍HEa₎miRNA mimic–induced reduction of *SNAI1* and *VIM* expression (post hoc Tukey's HSD, n = 10 samples per group, $P = 0.007$ and $P < 0.0001$, respectively) (Fig 6A–D).

As with cytotrophoblasts, pre-transfection with ₍HEa₎miRNA antagomirs prevented ₍HEa₎miRNA mimic–induced reduction of *VIM*, *SNAI1*, and *SNAI2* expression in extravillous trophoblasts (post hoc Tukey's

HSD, n = 10 samples per group, $P < 0.0001$, Fig 6E–H). Thus, our data suggest that pretreating cells with ₍HEa₎miRNA antagomirs prevents inhibition of EMT pathway members resulting from transfection with ₍HEa₎miRNA mimics in cytotrophoblasts and extravillous trophoblasts.

## ₍HEa₎miRNAs impair extravillous trophoblast invasion

Functionally, inhibition of the EMT pathway should reduce trophoblast invasiveness. Thus, we performed a transwell invasion assay using HTR8 extravillous trophoblasts transfected with ₍HEa₎miRNA mimics and antagomirs. Although ethanol exposure by itself did not impair trophoblast invasion (Fig S6), there was a marginally significant interaction effect between ethanol exposure and ₍HEa₎miRNA mimic transfection ($F_{(1,28)} = 3.418$, $P = 0.075$). Thus, a planned

comparison indicated that transfection with ₍HEa₎miRNA mimics significantly reduced trophoblast invasion in the context of 320 mg/dl ethanol co-exposure, relative to the control mimics (t(14) = 2.762, $P = 0.015$), consistent with our data demonstrating ₍HEa₎miRNAs interfere with the EMT pathway (Fig 7A). Contrastingly, transfecting ₍HEa₎miRNA antagomirs increased invasion in the context of 320 mg/dl ethanol co-exposure, although this effect was only marginally significant (t(14) = 1.805, $P = 0.093$, Fig 7B).

### ₍HEa₎miRNAs retard trophoblast cell cycle progression

Given the proliferative nature of cytotrophoblasts and the intimate relationship between EMT and cell cycle (64, 65), we assessed the

effects of ethanol and ₍HEa₎miRNAs on BeWO cytotrophoblast cell cycle. After pulse-labeling the cells with the nucleic acid analog, EdU, for 1-h, we found that individually transfecting six of the ₍HEa₎miRNA mimics increased EdU incorporation (unpaired t test, $P < 0.05$, false discovery rate [FDR] correction), suggesting an overall increased rate of DNA synthesis (Fig S7A). Contrastingly, simultaneous transfection of ₍HEa₎miRNAs significantly reduced EdU incorporation ($F_{(1,26)} = 59.69$, $P < 0.0001$), mirroring the effects of increasing concentrations of ethanol ($R^2 = 0.304$, $P = 0.012$) (Fig S7B and A).

Consistent with the increased rates of DNA synthesis resulting from individual ₍HEa₎miRNA mimic transfection, individual transfection of ₍HEa₎miRNAs antagomirs generally reduced EdU incorporation, although only the antagomir to hsa-miR-760 did so

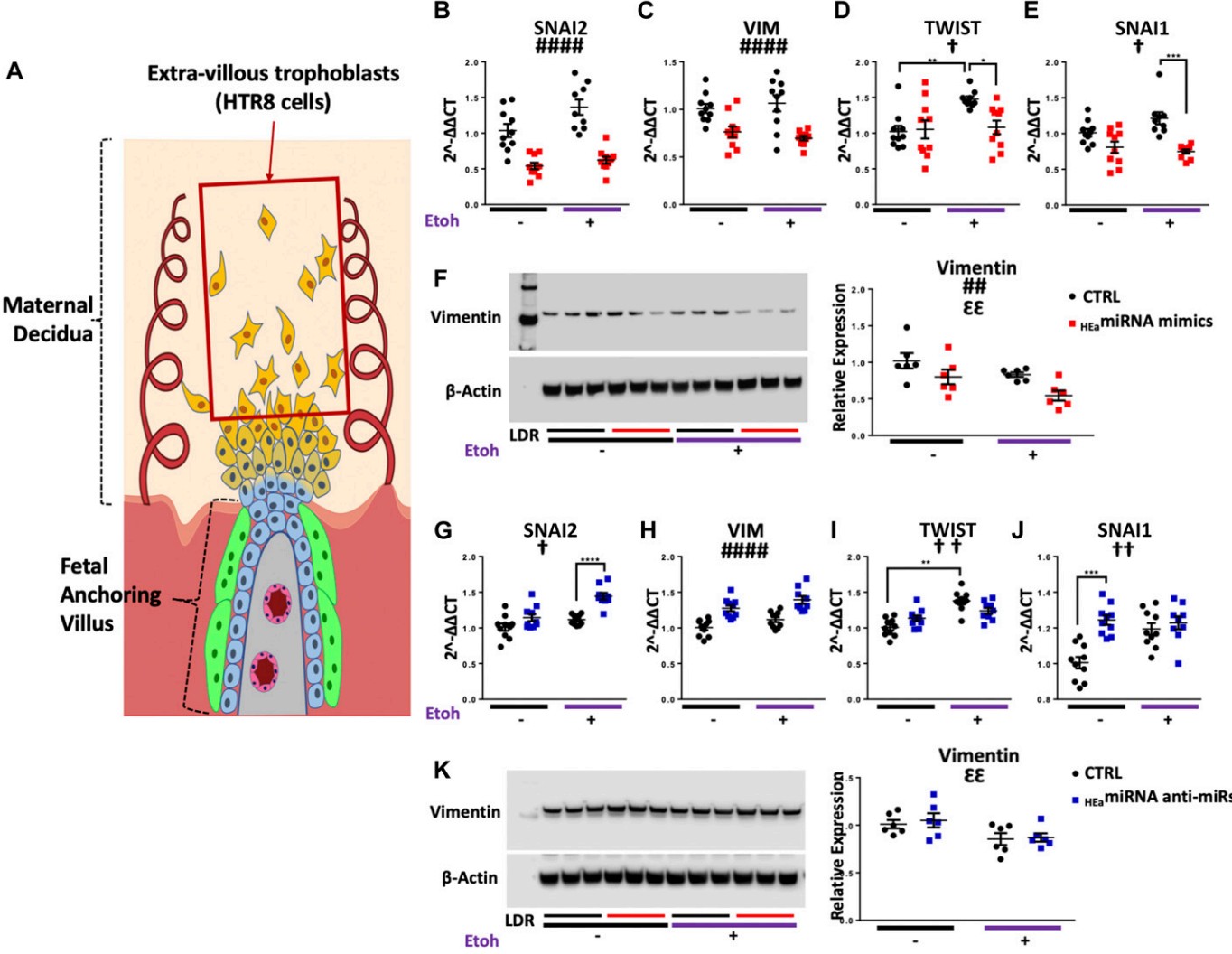

**Figure 5.** ₍HEa₎miRNAs interfere with the EMT pathway in HTR8 extravillous trophoblasts.
**(A)** Diagram of a placental anchoring villous and maternal decidua with the boxed area denoting extravillous trophoblasts. **(B–F)** Expression of *SNAI2* (B), *VIM* (C), *TWIST* (D), and *SNAI1* transcripts (E) as well as densitometric quantification of vimentin protein levels (F) in HTR8 extravillous trophoblasts following ₍HEa₎miRNAs or control miRNA overexpression with or without concomitant 320 mg/dl ethanol exposure. **(G–K)** Expression of *SNAI2* (G), *VIM* (H), *TWIST* (I), and *SNAI1* transcripts (J) as well as densitometric quantification of vimentin protein levels (K) in HTR8 extravillous trophoblasts following ₍HEa₎miRNA or control hairpin inhibitor transfection with or without concomitant 320 mg/dl ethanol exposure. Results are expressed as the mean ± SEM, LDR = molecular weight ladder, n = 10 samples per group; ANOVA: significant main effect of ₍HEa₎miRNA transfection (##$P < 0.01$, ####$P < 0.0001$), significant main effect of 320 mg/dl ethanol exposure (εε$P < 0.01$), significant interaction effect (₍HEa₎miRNA by 320 mg/dl ethanol (†$P < 0.05$, ††$P < 0.01$). For post hoc analysis *$P < 0.05$, **$P < 0.01$, ***$P < 0.001$, and ***$P < 0.0001$ by Tukey's HSD.

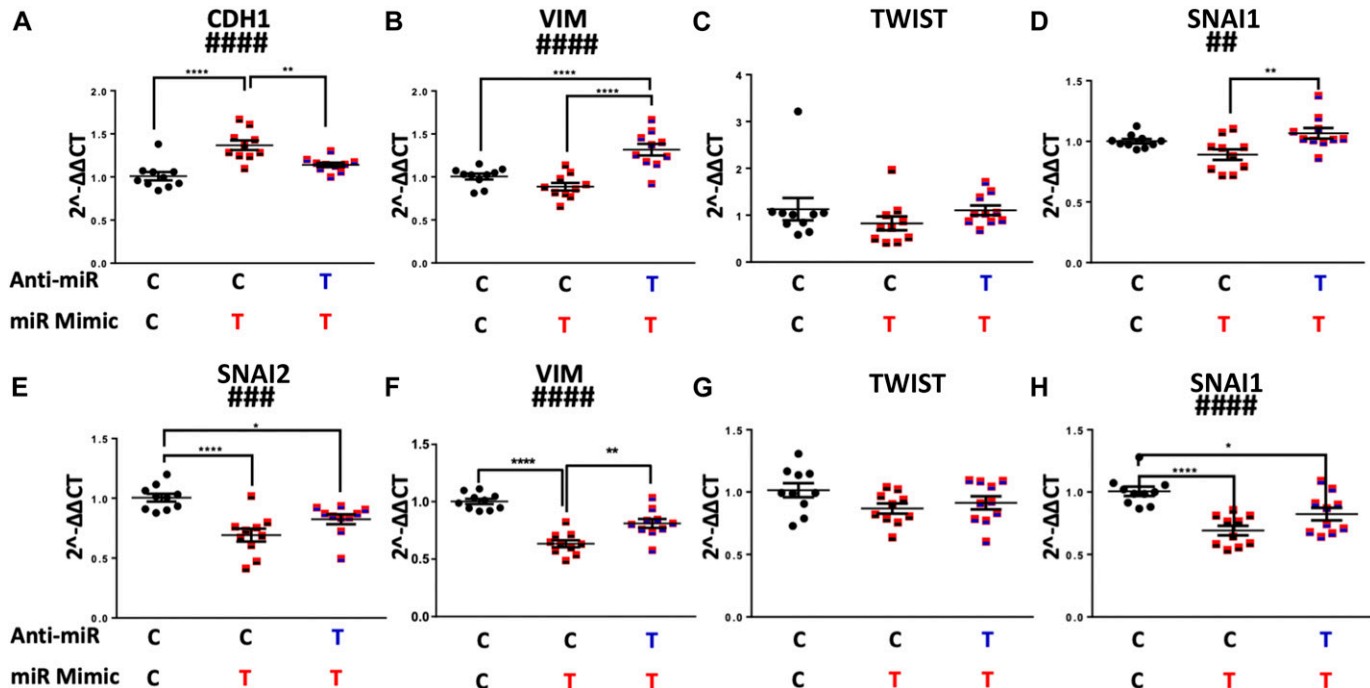

**Figure 6. Antagomirs prevent HEamiRNA-induced impairment of EMT.**
**(A–D)** Expression of *CDH1* (A), *VIM* (B), *TWIST* (C), and *SNAI1* transcripts (D) following control or HEamiRNA hairpin inhibitor transfection followed by control or HEamiRNA overexpression in BeWO cytotrophoblasts. **(E–H)** Expression of *CDH1* (E), *VIM* (F), *TWIST* (G), and *SNAI1* transcripts (H) following control or HEamiRNA antagomir transfection followed by control or HEamiRNA overexpression in HTR8 extravillous trophoblasts. In subheadings, 'C' denotes control miRNA mimic or hairpin, whereas 'T' denotes HEamiRNA mimic or hairpin inhibitor. Results are expressed as the mean ± SEM, n = 10 samples per group; ANOVA: significant treatment effect ($^{##}P < 0.01$, $^{###}P < 0.001$, $^{####}P < 0.0001$). For post hoc analysis, $^*P < 0.05$, $^{**}P < 0.01$, $^{***}P < 0.001$, $^{****}P < 0.0001$ by Tukey's HSD.

significantly (t(110) = 3.059, $P = 0.003$, FDR correction) (Fig S7A). Interestingly, simultaneous administration of antagomirs also reduced EdU incorporation, as observed with the pooled HEamiRNA mimics ($F_{(1,26)} = 34.83$, $P = 0.0005$, Fig 8B).

To further characterize the coordinated effect of HEamiRNAs on cytotrophoblast cell cycle, we pulse-labeled the cells with EdU for 1-h and, post-fixation, labeled them with 7AAD to segregate cells into three groups: $G_0/G_1$ (7AADlow, EDU–), S (EDU+), and $G_2/M$ (7AADhigh, EDU–). Both 120 mg/dl and 320 mg/dl ethanol exposures significantly decreased the proportion of cells in the S-phase, whereas 320 mg/dl exposure increased the proportion of cells in the $G_2/M$-phase, consistent with the observed reduction in the rate

of DNA synthesis (Fig S7C). Similar to the effects of ethanol exposure, pooled HEamiRNA mimic administration also significantly decreased the proportion of cells in the S-phase ($F_{(1,28)} = 52.78$, $P < 0.0001$), whereas increasing the proportion of cells the $G_2/M$-phase ($F_{(1,28)} = 8.395$, $P = 0.007$) and exacerbated alcohol's effects on the cell cycle (Fig 8C). Interestingly, pooled HEamiRNA antagomir administration also reduced the proportion of cells in the S-phase ($F_{(1,26)} = 14.98$, $P = 0.0007$) and increased the proportion of those in the $G_2/M$-phase ($F_{(1,26)} = 12.38$, $P = 0.002$) (Fig 8D).

As with our EMT gene expression data, pretreatment of cytotrophoblasts with HEamiRNA antagomirs prevented further reduction in the rate of DNA synthesis, or cell cycle retardation, that

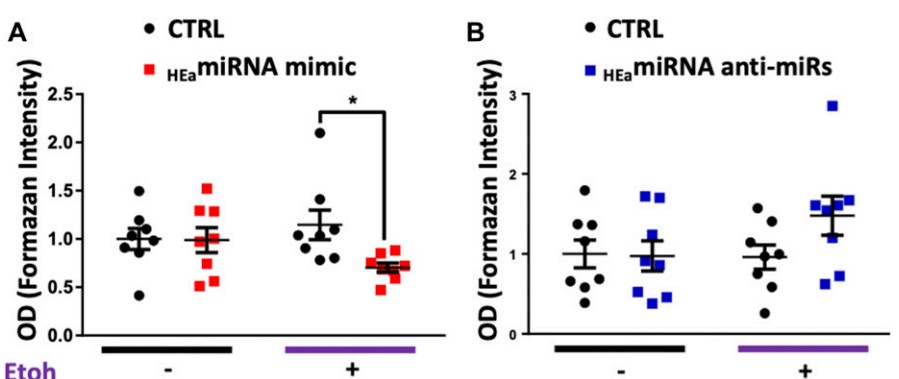

**Figure 7. HEamiRNAs impair extravillous trophoblast invasion.**
**(A, B)** Transwell invasion of HTR8 extravillous trophoblasts following transfection with (A) HEamiRNA mimics or (B) hairpin inhibitors with or without concomitant 320 mg/dl ethanol exposure. OD = optical density; results are expressed as the mean ± SEM; n = 10 samples per group; $^*P < 0.05$ by unpaired t test.

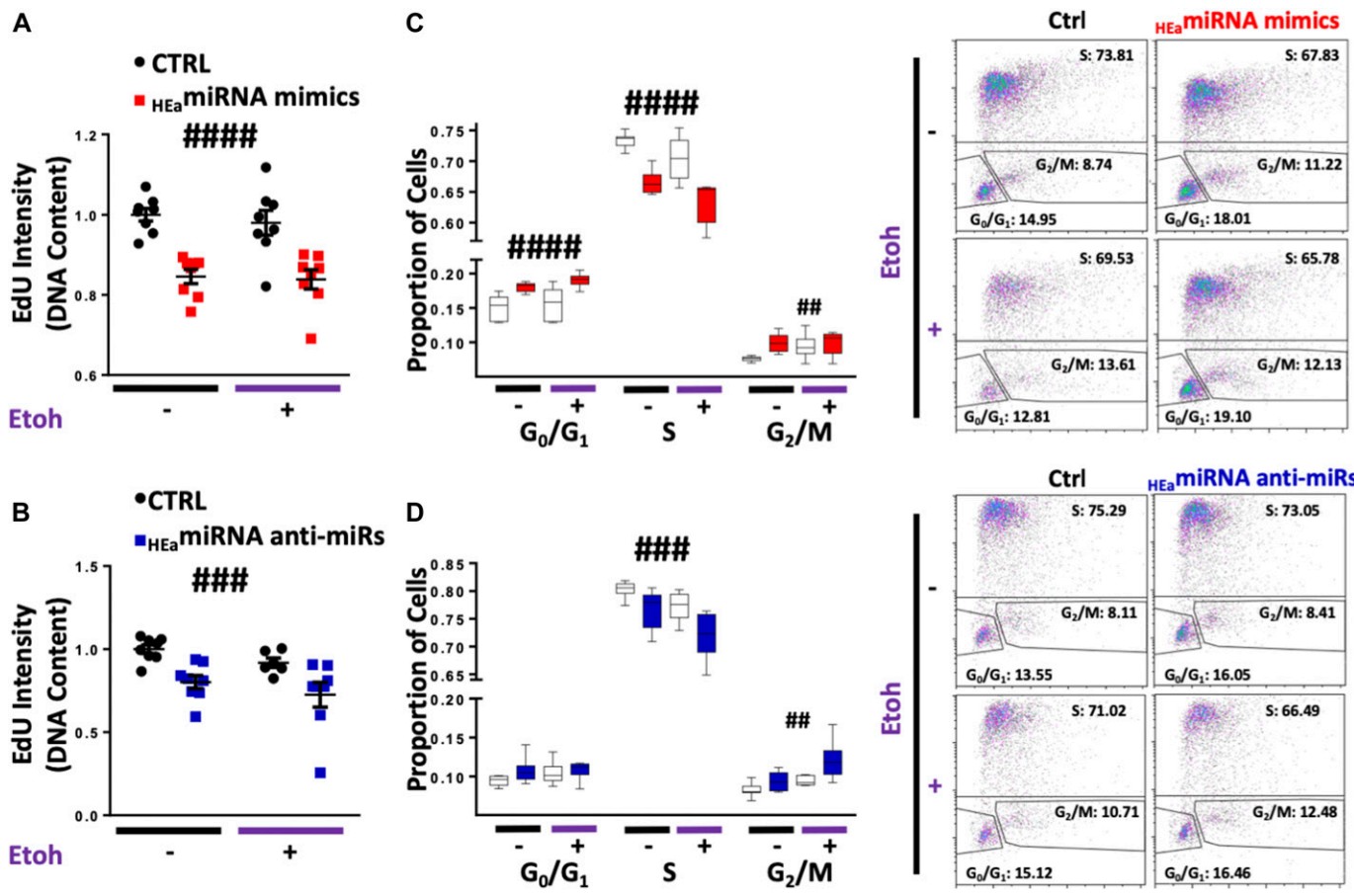

**Figure 8. HEamiRNAs cause cell cycle retardation in trophoblasts.**
**(A)** Degree of EdU incorporation following control and HEamiRNA overexpression. **(B)** Degree of EdU incorporation following control and HEamiRNA hairpin inhibitor transfection. **(C)** Box and whisker plot for the proportion of cells in the $G_0/G_1$, S, or $G_2/M$ phase of the cell cycle following control and HEamiRNA overexpression. **(D)** Box and whisker plot for the proportion of cells in the $G_0/G_1$, S, or $G_2/M$ phase of the cell cycle following control and HEamiRNA hairpin inhibitor transfection with or without concomitant 320 mg/dl ethanol exposure. For box and whisker plots, bounds of box demarcate limits of the first and third quartile, the line in middle is the median, and whiskers represent the range of data. Representative flow cytometry experiment images are shown on the right. n = 10 samples per group; ANOVA: significant main effect of HEamiRNA transfection ($^{##}P < 0.01$, $^{###}P < 0.001$, and $^{####}P < 0.0001$).

would result from transfection with pooled HEamiRNA mimics (Fig 9A and B).

## HEamiRNAs have minimal effect on cell survival

We next investigated whether ethanol- and HEamiRNA-induced changes in cell cycle were related to an increase in cell death. Only the 320 mg/dl dose of ethanol exposure demonstrated a slight, but marginally significant effect, of increasing lytic cell death (t(18) = 2.022, $P = 0.054$), although there was no effect on apoptosis (Fig S8A and B). However, the changes in cell cycle following transfection of individual or pooled HEamiRNA mimics were not mirrored by changes in lytic cell death. Nevertheless, two HEamiRNAs, hsa-miR-671-5p and hsa-miR-449a, did significantly increase apoptosis (unpaired t test, $P < 0.05$, FDR correction) (Fig S8C and D).

Contrastingly, transfection of four HEamiRNA antagomirs individually, significantly increased lytic cell death (unpaired t test, all $P < 0.05$, FDR correction), with the antagomir to hsa-miR-491-3p also increasing apoptotic cell death (t(14) = 3.383, $P = 0.004$, FDR

correction, Fig S8C and D). Likewise, transfection of pooled HEamiRNA antagomirs increased lytic cell death ($F_{(1,36)} = 11.40$, $P = 0.002$) but did not cause increased apoptosis (Fig S8E–H). Taken together, our data suggest that whereas ethanol exposure may increase cytotrophoblast death, increased levels of HEamiRNAs have minimal effects on cell death, suggesting that their effect on cell cycle and the EMT pathway is independent of any effect on cell survival.

## HEamiRNAs modulate cytotrophoblast differentiation-associated $Ca^{2+}$ dynamics

HEamiRNAs' effects on EMT pathway member expression, coupled with cell cycle retardation, indicate that HEamiRNAs influence trophoblast maturation. To model HEamiRNAs' effect on hormone-producing and calcium-transporting syncytiotrophoblasts (66), we used a well-established protocol of forskolin-induced syncytialization of BeWO cytotrophoblasts (67, 68). As expected, forskolin treatment induced fusion/syncytialization of cytotrophoblasts resulting in a greater average cell size in the forskolin + HEamiRNA

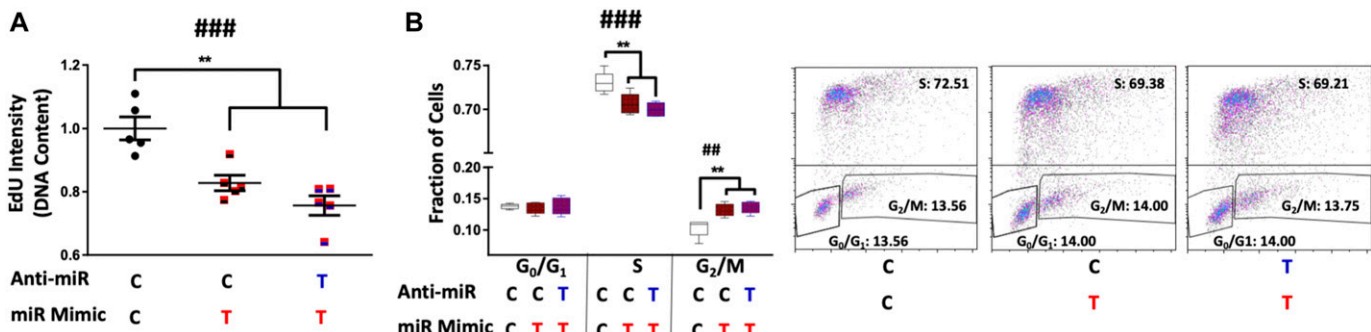

**Figure 9. Antagomirs prevent $_{HEa}$miRNA-induced cell cycle retardation.**
**(A)** Degree of EdU incorporation following control or $_{HEa}$miRNA hairpin inhibitor transfection followed by control or $_{HEa}$miRNA overexpression in BeWO cytotrophoblasts. Results are expressed as the mean ± SEM. **(B)** Box and whisker plot for the proportion of cells in the $G_0/G_1$, S, or $G_2/M$ phase of the cell cycle following control or $_{HEa}$miRNA hairpin inhibitor transfection followed by control or $_{HEa}$miRNA overexpression in BeWO cytotrophoblasts. Bounds of box demarcate limits of the first and third quartile, the line in middle is the median, and whiskers represent the range of data. Representative flow cytometry experiment images are shown on the right. In subheadings, 'C' denotes control miRNA mimic or hairpin, whereas 'T' denotes $_{HEa}$miRNA mimic or hairpin inhibitor. n = 5 samples per group; ANOVA: significant treatment effect ($^{###}P < 0.001$). For post hoc analysis, $^{**}P < 0.01$ by Tukey's HSD.

mimics group ($F_{(1,386)} = 4.386$, $P = 0.037$). This suggests that the inhibition of EMT by these miRNAs may result in preferential syncytialization instead of differentiation to extravillous trophoblasts (Fig S9A). Ethanol and forskolin treatment both increased baseline calcium levels, as indicated by the change in fluo-4 fluorescence ($F_{(1,426)} = 5.593$ and 3.665, respectively, $P < 0.0001$, Figs 10A and S9B–D).

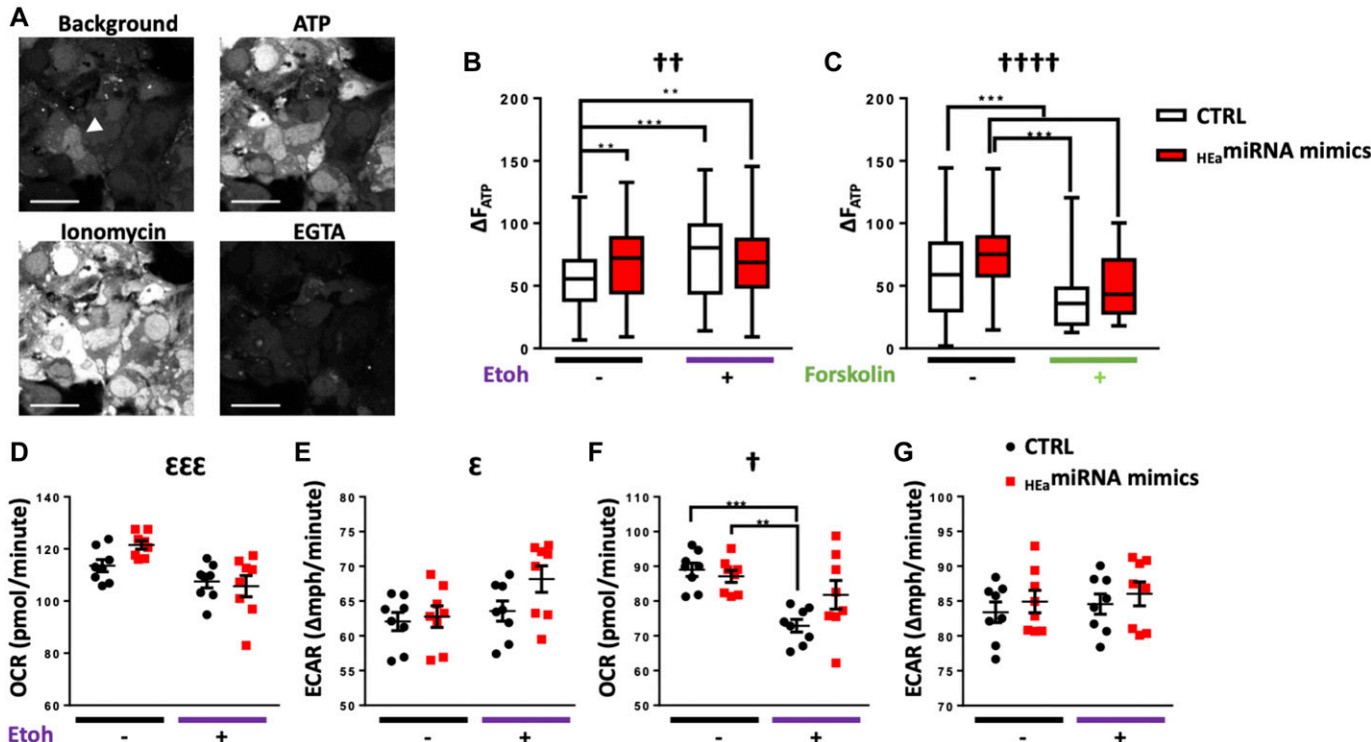

**Figure 10. $_{HEa}$miRNAs modulate differentiation-associated Ca$^{2+}$ dynamics but have minimal effect on the cellular energetics profile.**
**(A)** Time-lapse confocal images of BeWO cytotrophoblasts loaded with fluo-4 Ca$^{2+}$ indicator dye under indicated treatment conditions. The arrowhead indicates a fused, multinuclear cell, scale bar (white) is 50 $\mu$m. **(B)** Box and whisker plot of intracellular calcium levels following acute ATP administration in BeWO cytotrophoblasts with control and $_{HEa}$miRNA overexpression with or without concomitant 320 mg/dl ethanol exposure. Bounds of box demarcate limits of the first and third quartile, the line in middle is the median, and whiskers represent the range of data. **(C)** Box and whisker plot of intracellular calcium levels following acute ATP administration in BeWO cytotrophoblasts with control and $_{HEa}$miRNA overexpression with or without 20 $\mu$M forskolin treatment. **(D–G)** Baseline OCR (D), baseline ECAR (E), stressed OCR (F), and stressed ECAR (G) in BeWO cytotrophoblasts with control and $_{HEa}$miRNA overexpression with or without concomitant 320 mg/dl ethanol exposure. Metabolic stress was induced by treatment with 1 $\mu$M oligomycin and 0.125 $\mu$M (FCCP). Results are expressed as the mean ± SEM. n = 10 samples per group; ANOVA: significant main effect of 320 mg/dl ethanol exposure ($^{\varepsilon}P < 0.05$, $^{\varepsilon\varepsilon\varepsilon}P < 0.001$), significant interaction effect ($_{HEa}$miRNA by 320 mg/dl ethanol [$^{†}P < 0.05$, $^{††}P < 0.01$, and $^{††††}P < 0.0001$]). For post hoc analysis, $^{*}P < 0.05$, $^{**}P < 0.01$, $^{***}P < 0.001$, and $^{***}P < 0.0001$ by Tukey's HSD.

The effect of ethanol on baseline calcium was abrogated by $_{HEa}$miRNAs, whereas $_{HEa}$miRNAs + forskolin was not significantly different to forskolin alone, indicating that forskolin and $_{HEa}$miRNAs may be affecting similar calcium pathways. The conversion of cytrophoblasts to syncytiotrophoblasts is accompanied by an increase in endoplasmic reticulum, which could increase calcium-buffering capabilities in response to ethanol stress on the cells; thus, $_{HEa}$miRNA-induced syncytialization pathways may be protective against ethanol stress.

Adaptations to cellular stress can also be seen in alterations to cellular energetics in response to ethanol, as ethanol-exposed BeWO cells showed decreased baseline and stressed oxygen consumption rates (OCR) ($F_{(1,28)}$ = 15.55 and 16.91, $P$ = 0.0005 and 0.0003, respectively) and increased extracellular acidification rates

(ECAR) ($F_{(1,28)}$ = 4.868, $P$ = 0.036). However, $_{HEa}$miRNAs had minimal effects on metabolic activity (Fig 10D–G).

Extracellular ATP has been shown to inhibit trophoblast migration (69) and can directly stimulate increased intracellular calcium elevations through purinergic receptors ubiquitously present on trophoblasts (70). Both $_{HEa}$miRNA and ethanol administration significantly increased intracellular calcium in response to acute ATP administration ($F_{(1,426)}$ = 10.34 and $F_{(1,386)}$ = 16.30, $P$ = 0.001 and $P$ < 0.0001, respectively) (Fig 10B). This may be indicative of a lack of down-regulation of purinergic receptors required in trophoblast migration as part of the interrupted EMT pathway. Forskolin-induced maturation decreased calcium response to ATP ($F_{(1,386)}$ = 50.72, $P$ < 0.0001) (Fig 10C) and prevented the $_{HEa}$miRNA-induced increase in ATP response. These data agree with previous studies showing

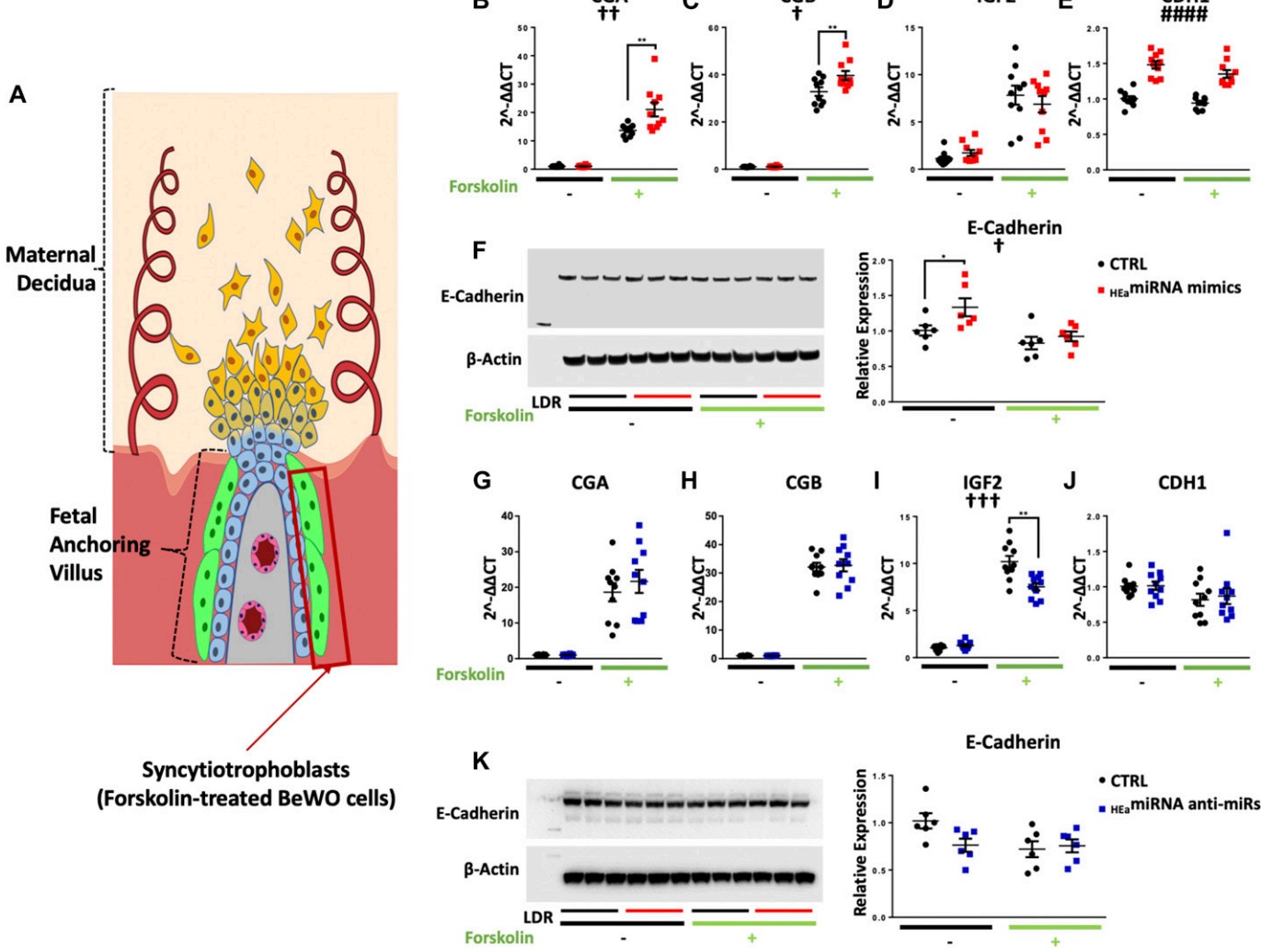

**Figure 11.** $_{HEa}$miRNAs promote syncytialization dependent hCG production.
**(A)** Diagram of a placental anchoring villous and maternal decidua with the boxed area denoting syncytiotrophoblasts. **(B–F)** Expression of *CGA* (B), *CGB* (C), *IGF2* (D), and *CDH1* transcripts (E) and densitometric quantification of E-Cadherin protein levels (F) in BeWO cytotrophoblasts following $_{HEa}$miRNAs or control miRNA overexpression with or without 20 *μM* forskolin treatment. **(G–K)** Expression of *CGA* (G), *CGB* (H), *IGF2* (I), and *CDH1* transcripts (J) and densitometric quantification of E-Cadherin protein levels (K) in BeWO cytotrophoblasts following $_{HEa}$miRNAs or control hairpin inhibitor transfection with or without 20 *μM* forskolin treatment. Results are expressed as the mean ± SEM, LDR = molecular weight ladder, n = 10 samples per group; ANOVA: significant main effect of $_{HEa}$miRNA transfection ($^{####}P$ < 0.0001), significant interaction effect ($_{HEa}$miRNA by forskolin, [$^{†}P$ < 0.05]). For post hoc analysis, $^{*}P$ < 0.05, $^{**}P$ < 0.01 by Tukey's HSD.

increased nuclear trafficking of ionotropic receptor P2X7 and more localized P2X4 expression over placental development, which may decrease the overall calcium influx in response to ATP (71).

### $_{HEa}$miRNAs promote syncytialization-dependent hormone production

Transfection of $_{HEa}$miRNA mimics did not change *CGA* (encodes Chorionic gonatropin alpha), *CGB* (encodes Chorionic gonadotropin beta), or *IGF2* (encodes Insulin-like growth factor 2) transcript expression relative to the control in non-syncytialized trophoblasts. However, following forskolin-induced syncytialization of BeWO cytotrophoblasts (Fig 11A), $_{HEa}$miRNA mimics significantly increased expression of *CGA* and *CGB* (post hoc Tukey's HSD, n = 10 samples per group, *P* = 0.001 and 0.005, respectively). Consistent with our previous results, $_{HEa}$miRNA mimics also increased *CDH1* expression in both cytotrophoblasts and syncytiotrophoblasts ($F_{(1,20)}$ = 5.286, *P* = 0.032); there was also a main effect of syncytialization on *CDH1* expression, as has been previously reported ($F_{(1,36)}$ = 3.391, *P* = 0.034, Fig 11B–E). Likewise, $_{HEa}$miRNAs increased E-Cadherin protein expression ($F_{(1,20)}$ = 5.286, *P* = 0.032), whereas forskolin decreased it ($F_{(1,20)}$ = 10.24, *P* = 0.005) (Fig 11F). On the other hand, there was no effect of $_{HEa}$miRNA antagomirs on *CGA* and *CGB* expression, although we did observe a decrease in *IGF2* transcript expression, following syncytialization, relative to controls (post hoc Tukey's HSD, n = 10 samples per group, *P* = 0.001) (Fig 11G–J).

Given that $_{HEa}$miRNAs promote syncytialization-dependent hormone production, we next investigated maternal plasma levels of intact hCG in our Ukraine birth cohort. Plasma hCG levels were nonsignificantly increased in the second trimester of HEa group mothers relative to their UE counterparts, consistent with previous studies (72). During the third trimester, however, hCG levels remained significantly elevated in HEa group mothers compared with the UE group (median test, n = 23 samples in HEa

group and n = 22 for HEua and UE groups, *P* = 0.03) (Fig 12). Furthermore, there was no significant difference of gestational age at blood draw between the different groups indicating the increased level of hCG in the HEa group was not confounded by gestational age at which blood was sampled (Fig S10) (73). Interestingly, both alcohol and hCG levels were negatively associated with gestational age at delivery (GAD), with a significant interaction between periconceptional alcohol exposure and hCG levels on GAD (Table S2). Taken together, our data suggest $_{HEa}$miRNAs may contribute to PAE-dependent increases in hCG levels during pregnancy.

### $_{HEa}$miRNAs reduce fetal growth

To investigate the functional consequences of elevated circulating $_{HEa}$miRNA levels, we administered miRNA mimics for the eight-mouse homolog $_{HEa}$miRNAs, or a negative control mimic, through tail vein injection to pregnant mouse dams on GD10. On GD18, growth parameters of male and female fetuses were assessed separately, and data from all same-sex fetuses from a single pregnancy were averaged into one data point. Dams-administered $_{HEa}$miRNA mimics produced smaller fetuses than those administered control mimics, according to all collected measures of fetal size: fetal weight ($F_{(1,17)}$ = 9.92, *P* = 0.006), crown-rump length ($F_{(1,17)}$ = 9.89, *P* = 0.006), snout-occipital distance ($F_{(1,17)}$ = 9.09, *P* = 0.008), and biparietal diameter ($F_{(1,17)}$ = 5.99, *P* = 0.026) (Fig 13B–E). Interestingly, placental weights were also significantly reduced in mice treated with $_{HEa}$miRNA mimics ($F_{(1,17)}$ = 6.92, *P* = 0.018) (Fig 13F).

Following tail vein administration of two human-specific sentinel miRNAs, miR-518f-3p and miR-519a-3p, we found a high biodistribution of both miRNAs in the placenta, comparable with levels seen in the liver and spleen (Fig S11A and B). Thus, to determine whether $_{HEa}$miRNA's effects on fetal growth could result from their actions on the placenta, we quantified the placental expression of core EMT members in the GD18 placentas of control and $_{HEa}$miRNA fetuses. $_{HEa}$miRNA administration significantly reduced expression of mesenchymal-associated transcript *VIM* ($F_{(1,14)}$ = 14.23, *P* = 0.002) and S*NAI2* ($F_{(1,14)}$ = 5.99, *P* = 0.028) with a significant sex by $_{HEa}$miRNA interaction effect on *SNAI1* ($F_{(1,66)}$ = 5.55, *P* = 0.034) and *CDH1* ($F_{(1,14)}$ = 6.01, *P* = 0.028) (Fig 14A–E). Interestingly, and in line with our in vitro findings whereby $_{HEa}$miRNAs promoted syncytialization-dependent cell fusion and hCG production, $_{HEa}$miRNA administration significantly increased expression of the mRNA transcript for *SynB*, a gene that is important for syncytiotrophoblast maturation ($F_{(1,66)}$ = 4.11, *P* = 0.047) (Fig 14F).

## Discussion

We previously reported that gestational elevation of 11 maternal plasma miRNAs predicted which PAE infants would exhibit adverse outcomes at birth (8). These $_{HEa}$miRNAs were elevated throughout mid and late-pregnancy, encompassing critical periods for fetal development, and were predicted to target the EMT pathway (8). In this study, we tested this prediction by adopting rodent and macaque gestational moderate alcohol self-administration paradigms. Despite differences in their placental anatomy (74, 75, 76, 77),

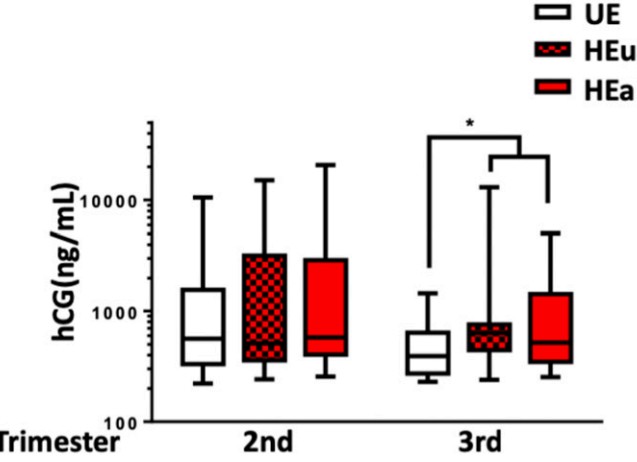

**Figure 12. PAE elevates third trimester maternal hCG.**
Box and whisker plot of the second and third trimester maternal hCG levels in UE, HEua, and HEa group mothers of our Ukrainian birth cohort. Bounds of box demarcate limits of the first and third quartile, the line in middle is the median, and whiskers represent the range of data. Results are expressed as the mean ± SEM, n = 22–23 samples per group; *P = 0.03 (Mood's median test, $\chi^2$ = 7.043, df = 2).

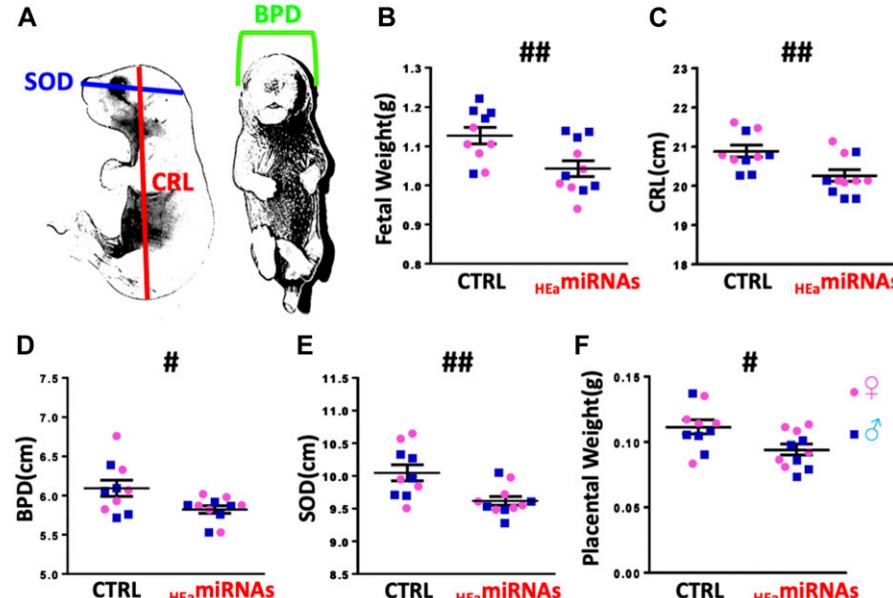

**Figure 13.** <sub>HEa</sub>miRNAs restrict fetal growth.
**(A)** Schematic for measures of crown rump length (CRL), biparietal diameter (BPD), and snout-occipital distance (SOD). **(B–F)** Fetal weight (B), crown-rump length (C), biparietal diameter (D), snout-occipital distance (E), and placental weight (F) at GD18 following administration of control (Ctrl) and <sub>HEa</sub>miRNA mimics to pregnant C57/Bl6 dams on GD10. Dots represent median measures of fetal size and placental weights from male and female offspring in independent litters. There were no significant differences in litter sizes (Ctrl: 8.2 and <sub>HEa</sub>miRNAs: 8.5) or sex ratios (Ctrl: 0.86 and <sub>HEa</sub>miRNAs: 1.21) between treatment conditions ($P > 0.5$ for all measures). Results are expressed as the mean ± SEM, n = 5–6 separate litters per treatment condition; ANOVA: significant main effect of <sub>HEa</sub>miRNA administration (#$P <$ 0.05 and ##$P < 0.01$).

we are the first to report that PAE impairs placental EMT across species, indicating a conserved effect of PAE on placental development. In addition, we found that <sub>HEa</sub>miRNAs collectively, but not individually, mediated the effects of PAE on core EMT pathway members and that, together, they inhibited EMT in human trophoblast culture models. Although we assessed the effects of <sub>HEa</sub>miRNAs on core EMT components (10, 14, 15, 59, 60, 61, 62), analysis of their 3' UTRs indicates that these are unlikely to be the direct targets of <sub>HEa</sub>miRNA action. Additional studies will be needed to dissect out the signaling networks that connect <sub>HEa</sub>miRNAs to the assessed EMT components.

Interestingly, <sub>HEa</sub>miRNAs also promoted syncytialization (forskolin)-dependent hCG expression, mirroring the elevation of third trimester

maternal hCG levels in the PAE group within our clinical cohort. This late-gestation elevation of hCG levels may serve as a compensatory mechanism to prevent the preterm birth associated with PAE, as hCG during late gestation is hypothesized to promote uterine myometrial quiescence (78, 79). In support of this hypothesis, we found significant negative associations between both hCG levels and alcohol consumption with GAD. Furthermore, there was a significant interaction between periconceptional alcohol exposure and hCG levels, with higher hCG levels corresponding to a smaller effect of alcohol exposure at conception on GAD, indicating that hCG moderates the effect of alcohol on age at delivery (Table S2).

Since <sub>HEa</sub>miRNAs collectively prevented trophoblast EMT, we hypothesized that, as a functional consequence, these maternal

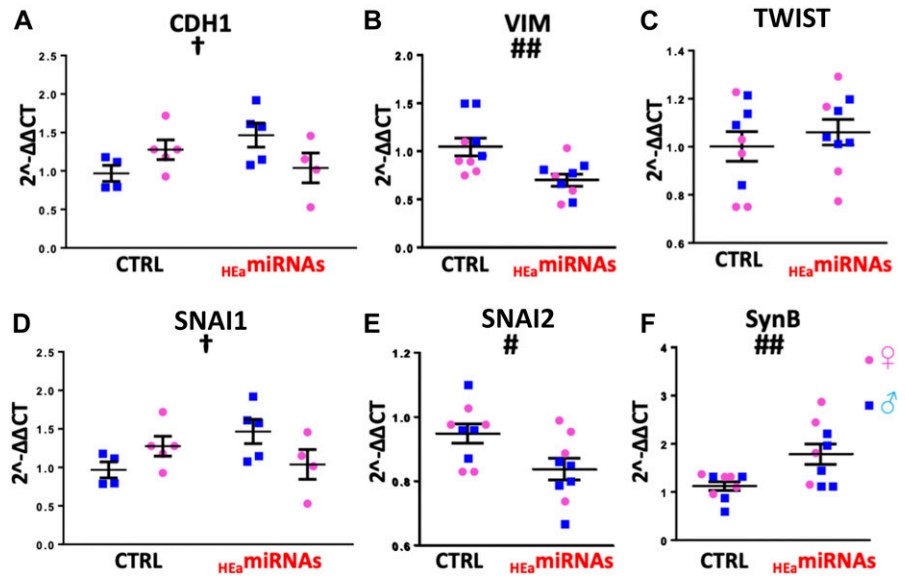

**Figure 14.** <sub>HEa</sub>miRNAs interfere with EMT in the placenta.
**(A–F)** Expression of *CDH1* (A), *VIM* (B), *TWIST* (C), *SNAI1* (D), and *SNAI2* (E) and *SynB* transcripts (F) in GD18 placenta following administration of control (Ctrl) and <sub>HEa</sub>miRNA mimics to pregnant C57/Bl6 dams on GD10. Dots represent median expression values of male and female offspring in independent litters. Results are expressed as the mean ± SEM, n = 5–6 separate litters per treatment condition, ANOVA: significant main effect of <sub>HEa</sub>miRNA administration (#$P < 0.05$, ##$P < 0.01$), significant interaction effect (fetal sex by <sub>HEa</sub>miRNA administration, [†$P < 0.05$]). For post hoc analysis, *$P < 0.05$ by Tukey's HSD.

miRNAs would also inhibit fetal growth. When we delivered 8 of the 11 HEamiRNAs known to be present in mouse, to pregnant dams during the period of placental branching morphogenesis and endometrial invasion, and when EMT is particularly active, we found that HEamiRNAs reduced fetal growth. Importantly, ethanol exposure during this period has also been shown to result in fetal growth deficits and dysmorphia in rodent PAE models (80, 81), suggesting that maternal miRNA-mediated deficits in trophoblast invasion may mediate some of the effects of PAE on fetal growth. In support of this, we found placentas from the HEamiRNA-treated group had impaired expression of core EMT pathway members. This disruption of placental EMT may also have implications for placental vascular dynamics, as we have also previously observed in mouse models (82). The nonhuman primate tissue analyzed here was also derived from animals that were characterized in vivo using MRI and ultrasound imaging, which demonstrated that maternal blood supply to the placenta was lower in ethanol-exposed animals compared with controls and that oxygen availability to the fetal vasculature was reduced (83).

HEamiRNAs may mediate other pregnancy-associated pathologies, aside from PAE. We identified numerous studies that reported increased circulating and placental levels of at least 8 of 11 HEamiRNAs in gestational pathologies arising from placental dysfunction. For example, elevated levels of one HEamiRNA, miR-519a-3p, a member of the placentally expressed C19MC family cluster, was reported in the placentae of patients with preeclampsia, recurrent spontaneous abortion, and intrauterine growth restriction (29, 30, 45, 46). Interestingly, collective overexpression of the 59 C19MC miRNAs inhibits trophoblast migration, explaining their enrichment in the non-migratory villous trophoblasts and suggests their down-regulation is necessary for maturation into invasive extravillous trophoblasts (84). Thus, a greater understanding of the placental roles of HEamiRNAs may also help disentangle the etiology of other pregnancy complications. We also observed that overexpression of more restricted subsets of HEamiRNAs associated with preeclampsia, fetal growth restriction, and spontaneous abortion or preterm labor also partly promoted EMT transcript signatures, contrasting with the collective inhibitory action of HEamiRNAs as a whole. Thus, elevation of some subsets of HEamiRNAs may constitute a compensatory mechanism aimed at minimizing placental pathologies, although their potential protective effects are masked by the collective elevation of HEamiRNAs.

Although we did not investigate the effects of PAE on EMT in nonplacental organs, it is likely that PAE broadly disrupts EMT in multiple fetal compartments. Developmental ethanol exposure has been shown to inhibit the EMT-dependent migration of neural crest progenitors involved in craniofacial development, explaining the facial dysmorphology seen in fetal alcohol syndrome and FASDs (85, 86). Outside of its effects on the neural crest, PAE is significantly associated with various congenital heart defects, including both septal defects and valvular malformations (87, 88, 89, 90). Given that development of heart depends on EMT within the endocardial cushions (91, 92), disruption of endocardial EMT could explain both the valvular and septal malformation associated with PAE.

Collectively, our data on HEamiRNAs suggest miRNA-based interventions could minimize or reverse developmental effects of PAE and other placental-related pathologies. miRNA-based therapeutic approaches have been advanced for other disease conditions (93, 94). However, our data also suggest the effects of combinations of miRNAs are not a sum of their individual effects. Functional synergy between clusters of co-regulated miRNAs may be a common feature in development and disease. For instance, in 2007, we presented early evidence that ethanol exposure reduced miR-335, -21, and -153 in neural progenitors and that coordinate reduction in these miRNAs yielded net resistance to apoptosis following ethanol exposure (95). In that study, we also showed that coordinate knockdown of these three miRNAs was required to induce mRNA for Jagged-1, a ligand for the Notch cell signaling pathway, an outcome that was not recapitulated by knocking down each miRNA individually (95). More recently, combined administration of miR-21 and miR-146a has been shown to be more effective in preserving cardiac function following myocardial infarction than administration of either of these miRNAs alone (96). Although miRNA synergy has not been explored in detail, these data show that new biology may emerge with admixtures of miRNAs and that therapeutic interventions may require the use of such miRNA admixtures rather than single miRNA molecules, as have been used in clinical studies to date.

In conclusion, we have observed that a set of 11 miRNAs, predictive of adverse infant outcomes following PAE, collectively mediate the effects of alcohol on the placenta. Specifically, elevated levels of these miRNAs together, but not individually, promote an aberrant maturational phenotype in trophoblasts by inhibiting core members of the EMT pathway and promoting cell stress and syncytialization-dependent hormone production. Although extensive research has established circulating miRNAs as biomarkers of disease, our study is one of the first to show how these miRNAs explain and control the disease process themselves. Functionally, we find that these miRNAs are clinically correlated with measures of fetal development and directly cause intrauterine growth restriction when administered in vivo. Our work suggests that a greater understanding for the role of HEamiRNAs during development, and their role in coordinating the EMT pathway in the placenta and other developing tissues, will benefit the understanding of FASDs and other gestational pathologies and potentially lead to effective avenues for intervention.

## Materials and Methods

### Mouse model of PAE

C57/BL6J mice (The Jackson Laboratory) were housed under reverse 12-h dark/12-h light cycle (lights off at 08:00 h). PAE was performed using a previously described limited access paradigm of maternal drinking (97, 98). Briefly, 60-d-old female mice were subjected to a ramp-up period with 0.066% saccharin containing 0% ethanol (2 d), 5% ethanol (2 d), and finally 10% ethanol for 4–h daily from 10:00 to 14:00 beginning 2 wk before pregnancy, continuing through gestation (Fig S2A). Female mice offered 0.066% saccharin without ethanol during the same time period throughout pregnancy served as controls. Tissue from the labyrinth, junctional, and decidual zone of male and female gestational day 14 (GD14) placentae were

microdissected, snap-frozen in liquid nitrogen, and stored at –80°C preceding RNA and protein isolation.

## Mouse model for HEamiRNA overexpression

For systemic administration of miRNAs, previously nulliparous C57/BL6NHsd dams (Envigo) were tail vein–injected on GD10 with either 50 μg of miRNA miRVana mimic negative control (Cat No. 4464061; Thermo Fisher Scientific) or pooled HEamiRNA miRVana mimics in In-vivo RNA-LANCEr II (3410-01; Bioo Scientific), according to the manufacturer's instructions. The 50 μg of pooled HEamiRNA mimics consisted of equimolar quantities of mmu-miR-222-5p, mmu-miR-187-5p, mmu-miR-299a, mmu-miR-491-3p, miR-760-3p, mmu-miR-671-3p, mmu-miR-449a-5p, and mmu-miR-204-5p mimics. For biodistribution studies, 50 μg of pooled equimolar quantities of hsa-miR-519a-3p and hsa-miR-518f-3p mimics were injected via tail vein. These human miRNAs were selected because no mouse homologs are known to exist and consequently, estimates for organ distribution of exogenous miRNAs in the mouse are unlikely to be contaminated by the expression of endogenous murine miRNAs. GD10 is a time point near the beginning of the developmental period of branching morphogenesis, immediately following chorioallantoic attachment, during which the placenta invades the maternal endometrium (99). At GD18, pregnancies were terminated with subsequent quantification of fetal weight, crown-rump length, snout-occipital distance, biparietal diameter, and placental weight (Fig 13A). Subsequently, tissue was snap-frozen in liquid nitrogen and stored at –80°C preceding RNA isolation.

## Rat model of PAE

Outbred nulliparous Sprague Dawley rats were housed under a 12-h light/12-hour dark cycle. PAE in Sprague Dawley was conducted according to our previously published exposure paradigm (20, 100). Briefly, dams were given a liquid diet containing either 0% or 12.5% ethanol (vol/vol) from 4 d before mating until GD4 (Fig S2B). Dams had ad libitum access to the liquid diet 21 h daily and consumed equivalent calories. Water was offered during the remaining 3 h of the day. On GD5, liquid diets were removed and replaced with standard laboratory chow. On GD20, the placentas were immediately separated into the labyrinth and junctional zone, snap-frozen in liquid nitrogen and stored at –80°C preceding RNA isolation.

## Nonhuman primate model of PAE

As previously described in detail (83), adult female rhesus macaques were trained to orally self-administer either 1.5 g/kg/d of 4% ethanol solution (equivalent to six drinks/d) or an isocaloric control fluid before time-mated breeding. Each pregnant animal continued ethanol exposure until gestational day 60 (GD60, term gestation is 168 d in the rhesus macaque) (101). Pregnancies were terminated by cesarean section delivery at three different time points; GD85, GD110, or GD135 (Fig S2C). The macaque placenta is typically bilobed with the umbilical cord insertion in the primary lobe and bridging vessels supplying the fetal side vasculature to the secondary lobe (Fig 2D showing gross placenta anatomy) (102). Full thickness tissue biopsies

(maternal decidua to fetal membranes) were taken from both the primary and secondary lobes of the placenta (Fig 2E showing H&E section of placenta). Samples were immediately snap-frozen in liquid nitrogen and stored at –80°C preceding RNA isolation.

## Cell culture trophoblast models

BeWO human cytotrophoblastic choriocarcinoma cells and HTR-8/SVneo extravillous cells were sourced from ATCC (Cat No. CCL-98 and CRL-3271, respectively). BeWO cells were maintained in HAM's F12 media containing penicillin (100 U/ml), streptomycin (100 μg/ml), and 10% vol/vol FCS at 37°C and 5% $CO_2$. HTR8 cells were maintained in RPMI-1640 media with 5% vol/vol FCS, under otherwise identical conditions. Culture medium was replenished every 2 d and cells subcultured every 4–5 d.

BeWO cells were treated with 20 μM forskolin to induce syncytialization, as previously described (103, 104). BeWO and HTR8 cells were also subjected to four separate ethanol treatment conditions: 0 mg/dl, 60 mg/dl (13 mM), 120 mg/dl (26 mM), or 320 mg/dl (70 mM). To achieve HEamiRNA overexpression and inhibition, Dharmacon miRIDIAN miRNA mimics and hairpin inhibitors (25 nM), or control mimic (Cat No. CN-001000-01-05; Dharmacon) and hairpin inhibitor (Cat No. CN-001000-01-05; Dharmacon) (25 nm), were transfected into subconfluent BeWO and HTR8 cells using RNAiMAX lipofection reagent (Cat No. 13778; Thermo Fisher Scientific).

## Cell cycle analysis

At 48 h post transfection, BeWO cells were pulsed with 10 μM EdU for 1 h. The cells were immediately harvested, and cell cycle analysis was performed with the Click-iT EdU Alexa Fluor 488 Flow Cytometry Assay kit (Cat No. C10420; Thermo Fisher Scientific), in conjunction with 7-amino-actinomycin D (Cat No. 00-6993-50; Thermo Fisher Scientific), according to the manufacturer's instructions, using the Beckman Coulter Gallios 2/5/3 flow cytometer. Data were analyzed using Kaluza software (Beckman Coulter).

## Cell death analysis

BeWO cell culture was harvested 48 h post transfection. Media was subjected to lactate dehydrogenase (LDH) detection using the Pierce LDH Cytotoxicity Assay kit (Cat No. 88953; Thermo Fisher Scientific), according to the manufacturer's instructions, for lytic cell death quantification. The Promega Caspase-Glo 3/7 Assay system (Cat No. G8091; Promega) was used to quantify apoptotic cell death.

## Invasion assay

At 24 h post-transfection and/or ethanol exposure, HTR8 cells were serum-starved for an additional 18 h. Subsequently, HTR8 cells were seeded onto transwell permeable supports precoated with 300 μg/ml Matrigel (Cat No. 354248; Corning). After 24 h, cells remaining in the apical chamber were removed with a cotton swab. Cells that invaded into the basal chamber were incubated with 1.2 mM 3-(4,5-dimethylthiazol-2-yl)-2,5-diphenyltetrazolium bromide (MTT) for 3 h, and the precipitate solubilized with 10% SDS in

0.01 N HCl. Absorbance intensities were read at 570 nm in a Tecan Infinite 200 plate reader.

## Metabolic flux analysis and calcium imaging

BeWO cells (10,000/well) were plated into Seahorse XF96 Cell Culture Microplates (Cat No. 103275-100; Agilent Biotechnology). The OCR, a measure of mitochondrial respiration, and ECAR, a measure of glycolysis, were measured using the Seahorse XFe96 flux analyzer (Seahorse Bioscience). At the time of assay, the cell culture medium was replaced with the appropriate prewarmed Seahorse XF Base Medium (Cat No. 102353-100; Agilent Biotechnology). OCR and ECAR parameters were measured using the Seahorse XFp Cell Energy Phenotype Test kit (Cat No. 103275-100; Agilent Biotechnology). Metabolic stress was induced by simultaneous treatment with 1 $\mu m$ oligomycin and 0.125 $\mu M$ carbonyl cyanide p-[trifluoromethoxy]-phenyl-hydrazone (FCCP).

BeWO cells were also plated onto glass coverslips in 24-well plates at a density of 30,000 cells/well. After exposure to ethanol and/or forskolin in culture, the cells were prepared for calcium imaging. After replacement of the culture media with external imaging media (154 mM NaCl, 5 mM KCl, 2 mM CaCl$_2$, 0.5 mM MgCl$_2$, 5 mM glucose, and 10 mM Hepes, pH 7.4), the cells were loaded for 35 min at 37°C with the calcium indicator dye fluo-4 AM (Cat No. F14201; Thermo Fisher Scientific), at a final concentration of 5 $\mu M$ fluo-4 AM in 0.1% DMSO. After incubation, the cells were washed to remove remaining extracellular fluo-4 and imaged at 40× using confocal microscopy (FV1200-equipped BX61WI microscope; Olympus Corporation). Time-lapse images were acquired at a frequency of 0.5 Hz. Individual cells were manually outlined, and area and mean fluorescence intensity were obtained for each cell (FIJI image processing package) [105]. To determine the functional calcium range of each cell, at the end of imaging, the cells were exposed to 5 $\mu M$ ionomycin and 10 mM EGTA (0 mM external Ca$^{2+}$, $F_{range} = F_{ionomycin} - F_{EGTA}$). Baseline fluorescence was determined by averaging the lowest five consecutive fluorescence values during the initial 5 min ($F_{baseline}$), which was then expressed as a percentage of $F_{range}$ ($\Delta F_{baseline} = (F_{baseline} - F_{EGTA})/F_{range} \times 100$). Maximal intracellular calcium response to 100 $\mu M$ ATP was determined by averaging the highest three consecutive fluorescence values during ATP application ($F_{ATP}$) and determining the amount of fluorescence as a percentage of $F_{range}$ ($\Delta F_{ATP} = (F_{ATP} - F_{EGTA})/F_{range} \times 100$).

## Quantitative reverse transcriptase–polymerase chain reaction analysis

Total RNA was extracted from tissue, as well as BeWO and HTR8 cells, using the miRNeasy Mini kit (Cat No. 217004; QIAGEN). For miRNA qPCR assays, cDNA was synthesized from 200 ng of total RNA using the miRCURY LNA Universal RT cDNA synthesis kit (Cat No. 203301; Exiqon/Cat No. 339340; QIAGEN), and expression was assessed using miRCURY LNA SYBR Green (Cat No. 203401; Exiqon/Cat No. 339345; QIAGEN). For mRNA qPCR assays, cDNA was synthesized from 500 ng of total RNA using the qScript cDNA Synthesis kit (Cat No. 95047; Quanta/QIAGEN). Gene expression analysis was performed using PerfeCTa SYBR Green FastMix (Cat No. 95073; Quanta) on the ViiA 7 Real-Time PCR System (Thermo Fisher Scientific). The data presented correspond to the mean $2^{-\Delta\Delta Ct}$ after being normalized to the geometric mean of $\beta$-actin, hypoxanthine-guanine phosphoribosyltransferase 1 (HPRT1), and 18s rRNA. Expression data for miRNA was normalized to the geometric mean of miR-25-3p, miR-574-3p, miR-30b-5p, miR-652-3p, and miR-15b-5p. For each primer pair, thermal stability curves were assessed for evidence of a single amplicon, and the length of each amplicon was verified using agarose gel electrophoresis. A list of primers and their sequences is presented in Table S3.

## Western immunoblotting analysis

Protein was extracted using 1× RIPA lysis buffer (MilliporeSigma) supplemented with Halt Protease Inhibitor Cocktail (Thermo Fisher Scientific). Tissue was homogenized using the Branson Sonifier 150. Protein concentration was determined using Pierce BCA protein assay kit (Thermo Fisher Scientific), and 30 $\mu g$ of protein was loaded onto a 4%–12% Bis-Tris (Cat No. NP0323BOX; Invitrogen/Thermo Fisher Scientific), size-fractionated at 200 V for 35 min, and transferred to a PVDF membrane using the iBlot transfer system (Invitrogen/Thermo Fisher Scientific). Blots with protein from cultured cells were blocked with 5% nonfat dry milk in tris-buffered saline containing Tween-20 (TTBS) for 1 h and incubated overnight with primary antibody. The blot was then washed and incubated with an HRP-conjugated goat anti-rabbit or anti-mouse IgG (Invitrogen) at dilution 1:1,000 for 1-h, then developed using PerkinElmer Western Lightning Plus Chemi ECL (PerkinElmer) and visualized using a CCD camera (Fluorchem Q, Alpha Innotech). Blots with protein from homogenized tissue were dried overnight, rehydrated in methanol, stained with REVERT Total Protein Stain, and developed with the Odyssey CLx Imaging System (LI-COR). Blots were then blocked with Odyssey Blocking Buffer (TBS) for 1 h and incubated overnight with primary antibody. The blot was then washed and incubated with IRDye 800CW secondary antibody (Cat No. 925-32210; LI-COR). The following antibodies were used: $\beta$-Actin HRP (Cat No. sc-47778; Santa Cruz Biotechnology); Goat anti-Mouse IgG (H+L) Secondary Antibody, HRP (Cat No. 62-6520; Thermo Fisher Scientific); Goat anti-Rabbit IgG (H+L) Secondary Antibody, HRP (Cat No. 65-6120; Thermo Fisher Scientific); purified Mouse Anti-E-Cadherin (Cat No. 610181; BD Biosciences); and Rabbit anti-vimentin antibody (EPR3776) (Cat No. ab 924647; Abcam). Protein levels were quantified using the densitometric analysis package in FIJI image processing software [105].

## ELISA

The second and third trimester maternal plasma samples were collected as part of a longitudinal cohort study conducted in two regions of Western Ukraine as part of the Collaborative Initiative on FASDs (CIFASD.org) between the years 2006 and 2011, as previously reported [8]. Plasma, at a 1:1,000 dilution, was subjected to hCG detection using Abcam's intact human hCG ELISA kit (Cat no. ab100533) following the manufacturer's protocol.

## Literature review

We conducted a literature review for $_{HEa}$miRNAs and their associated gestational pathology using the National Institute of Health's PubMed search interface. For each miRNA, the following search parameters were used:

[*miRX OR miR X OR miRNA X OR miRNAX or miRNX*]

*AND MeSH Term*

where X represents the miRNA of interest and automatic term expansion was enabled. The following MeSH terms, and related search terms (in brackets), were used: Fetal Growth Retardation (Intrauterine Growth Retardation, IUGR, Intrauterine Growth Restriction, Low Birth Weight, LBW, Small For Gestational Age, SGA), Premature Birth (Preterm Birth, Preterm Birth, Preterm Infant, Premature Infant, Preterm Labor, Premature Labor), Spontaneous Abortion (Early Pregnancy Loss, Miscarriage, Abortion, Tubal Abortion, Aborted Fetus), Pre-Eclampsia (Pre Eclampsia, Pre-eclampsia, Pregnancy Toxemia, Gestational Hypertension, Maternal Hypertension), and Maternal Exposure (Environmental Exposure, Prenatal Exposure). Returned articles were subsequently assessed for relevance.

## Secondary analysis of RNA sequencing data

Expression levels of $_{HEa}$miRNAs in tissues were determined using the Human miRNA Expression Database and the miRmine Human miRNA expression database (58, 106). For expression analysis of $_{HEa}$miRNA pri-miRNAs, RNA sequencing data were used from NCBI's sequence read archive (https://www.ncbi.nlm.nih.gov/sra). The accession numbers for the sequence files are uterus (SRR1957209), thyroid (SRR1957207), thymus (SRR1957206), stomach (SRR1957205), spleen (SRR1957203), small intestine (SRR1957202), skeletal muscle (SRR1957201), salivary gland (SRR1957200), placenta (SRR1957197), lung (SRR1957195), liver (SRR1957193), kidney (SRR1957192), heart (SRR1957191), whole brain (SRR1957183), adrenal gland (SRR1957124), bone marrow (ERR315396), colon (ERR315484), adipose tissue (ERR315332), and pancreas (ERR315479). Deep sequencing analysis was conducted using the Galaxy version 15.07 user interface according to the bioinformatics pipeline outlined in Fig S12.

## Statistical analyses

Linear regression models were used to estimate associations between infant growth measures and miRNA expression levels, gestational age at blood draw, the interaction between subject-centered miRNA expression level and gestational age at blood draw, and child sex. Spearman correlations between infant growth measures and subject-centered miRNA expression levels were also calculated. Linear regression models were also used to estimate the associations between gestational at birth and log-transformed hCG levels, ethanol intake, the interaction between log-transformed hCG levels and ethanol intake, gestational at blood draw, and child sex. Statistical analysis and graphs were generated with GraphPad Prism 6 software (GraphPad Software, Inc), SPSS v24, or R version 3.3.1. Results are expressed as the mean ± SEM or alternatively as box and whisker plots with the bounds of the box demarcating limits of first and third quartile, a median line in the center of the box, and whiskers representing the total range of data. The overall group effect was analyzed for significance using one-way MANOVA, one-way or two-way ANOVA with Tukey's HSD or Dunnett's multiple comparisons post hoc testing when appropriate (i.e., following a significant group effect in one-way ANOVA or given a significant interaction effect between experimental conditions in two-way ANOVA), to correct for a family-wise error rate. A two-tailed $t$ test was used for planned comparisons. For experiments characterizing the individual effects of $_{HEa}$miRNAs against the control miRNA or antagomirs, individual two-tailed $t$ test with 5% FDR correction was applied to account for multiple comparisons. All statistical tests, sample sizes, and post hoc analysis are appropriately reported in the results section. A value of $P < 0.05$ was considered statistically significant and a value of $0.1 < P < 0.05$ was considered marginally significant.

## Study approval

Human study protocols were approved by the institutional review boards at the Lviv National Medical University, Ukraine, and the University of California San Diego as well as Texas A&M University in the United States. Research was conducted according to the principles expressed in the Declaration of Helsinki with written informed consent received from participants before inclusion in the study. All rodent experiments were performed in accordance with protocols approved by the University of New Mexico Institutional Animal Care and Use Committee (IACUC), the Texas A&M University IACUC, and the University of Queensland Animal Ethics Committees. All procedures involving nonhuman primate research subjects were approved by the IACUC of the Oregon National Primate Research Center (ONPRC), and guidelines for humane animal care were followed. The ONPRC abides by the Animal Welfare Act and Regulations enforced by the US Department of Agriculture.

# Supplementary Information

# Acknowledgements

This research was supported by grants from the NIH, P50 AA022534 (AM Allan), U01 AA014835 and the Office of Dietary Supplements (CD Chambers), R24 AA019431 (KA Grant), R01 AA021981 (CD Kroenke), R01 AA024659 (RC Miranda), and F31 AA026505 (AM Tseng). We thank the National Health and Medical Research Council of Australia (KM Moritz) for their support. We thank CIFASD for intellectual support and Megan S Pope and Tenley E Lehman for their assistance in conducting cell culture and animal studies. Data on human subjects are deposited at CIFASD.org, in accordance with NIH data repository guidelines.

## Author Contributions

AM Tseng: investigation, methodology, and writing—original draft, review, and editing.
AH Mahnke: investigation, methodology, and writing—review and editing.
AB Wells: formal analysis and writing—review and editing.
NA Salem: methodology and writing—review and editing.
AM Allan: investigation, methodology, and writing—review and editing.
VHJ Roberts: investigation, methodology, and writing—review and editing.
N Newman: investigation, methodology, and writing—review and editing.
NAR Walter: investigation, methodology, and writing—review and editing.
CD Kroenke: investigation, methodology, and writing—review and editing.
KA Grant: investigation, methodology, and writing—review and editing.
LK Akison: investigation, methodology, and writing—review and editing.
KM Moritz: investigation, methodology, and writing—review and editing.
CD Chambers: supervision, investigation, methodology, and writing—review and editing.
RC Miranda: supervision, investigation, methodology, and writing—review and editing.

## Conflict of Interest Statement

The authors declare that they have no conflict of interest.

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
