## [Reviewer comments · Life Science Alliance]

Maternal Circulating MiRNAs That Predict Infant FASD Outcomes Influence Placental Maturation

Alexander Tseng, Amanda Mahnke, Alan Wells, Nihal Salem, Andrea Allan, Victoria Roberts, Natali Newman, Nicole Walter, Christopher Kroenke, Kathleen Grant, Lisa Akison, Karen Moritz, Christina Chambers, and Rajesh Miranda
DOI: 10.26508/lsa.201800252

Review timeline:	Submission Date:	2018-11-22
	Editorial Decision:	2019-01-16
	Appeal Received:	2019-01-16
	Editorial Decision:	2019-01-19
	Revision Received:	2019-02-09
	Editorial Decision:	2019-02-19
	Revision Received:	2019-02-20
	Accepted:	2019-02-21

Scientific Editor: Andrea Leibfried

Transaction Report:

January 16, 2019

Re: Life Science Alliance manuscript #LSA-2018-00252-T

Rajesh C Miranda
Texas A&M University Health Science Center

Dear Dr. Miranda,

Thank you for submitting your manuscript entitled "Maternal Circulating MiRNAs That Predict Infant FASD Outcomes Influence Placental Maturation" to Life Science Alliance. Please excuse the delay in getting back to you, the time to secure reviewers was exceptionally long in this case and we furthermore received one of the reports with a delay. Your manuscript has been seen by two expert reviewers, whose reports are appended below.

As you will see, while your work ultimately deserves publication, reviewer #1 points out that the value of your data to others is relatively low. We discussed your work within our editorial team in light of the input received from both reviewers. I am afraid we concluded, given our aim to publish work of significant value to others in the field, that we cannot offer to publish your manuscript here. We are thus returning your manuscript to you to enable efficient publication elsewhere. We'd be happy to forward the reviewer comments to a journal of your choice to allow rapid publication based on existing reviewer reports.

We are sorry our decision is not more positive, but hope that you find the reviews constructive. Of course, this decision does not imply any lack of interest in your work and we look forward to future submissions from your lab.

Thank you for your interest in Life Science Alliance.

Sincerely,

Andrea Leibfried, PhD
Executive Editor
Life Science Alliance
Meyershofstr. 1
69117 Heidelberg, Germany
t +49 6221 8891 502
e a.leibfried@life-science-alliance.org
www.life-science-alliance.org

Reviewer #1 (Comments to the Authors (Required)):

This is an interesting paper and should be published in some form. There do not appear to be any methodological or design issues and stats are appropriate. The slight lack of enthusiasm comes

from the fact that what the authors did was cross-check their human data (PLoSOne 2016) in rodent, non-human primate and in vitro, with a mini review added. The authors come to the same conclusions as from the human study. The utility of the additional findings have not been made clear - leaving the "so what" question for the publication not fully answered. The results are resented in a long-hand way with quite a discursive style and could do with organising better for the reader to follow. The quality of some of the figures in the pdf version I saw was not ideal - too small with sloppy font changes.

I do think that the authors could strengthen the manuscript with a clear address of the clinical relevance of the additional data.

Reviewer #2 (Comments to the Authors (Required)):

This paper focused on miRNAs, the expressions of which are significantly elevated in the circulation of pregnant mothers who have infants affected by prenatal alcohol exposure in their Ukrainian cohort. The overexpression of miRNAs disrupts placental development, suggesting that these miRNAs may contribute to the pathology of Fetal Alcohol Spectrum Disorders (FASD). The study used not only rodent and primate models but also in vitro models of human placental trophoblast development.

Although the direct targets of these miRNAs are still elusive, I think this manuscript indicates nicely that the set of miRNAs defined in human patients is potentially involved in pathogenesis in FASD. Their data will be foundation of the mechanisms of FASD placenta pathology and may lead to novel intervention that ameliorates FASD symptoms.

Major comment:

The manuscript showed overlapping miRNAs between FASD and other diseases such as preeclampsia, spontaneous abortion, fetal growth restriction etc. Since impairment of trophoblast EMT is foundation of these diseases and the 11 HEamiRNAs collectively prevent this process, I wonder if subsets of HEamiRNAs shared between FASD and other diseases affect placental trophoblast development in vitro. Can authors add comments on it?

We thank you for the review of this manuscript, but we respectfully ask for a re-assessment of your decision to reject.

Neither reviewer found structural faults with the paper. However, we believe that Reviewer #1 made an error in assessing that , "what the authors did was cross-check their human data (PLoSOne 2016) in rodent, non-human primate and in vitro, with a mini review added", a statement which appears to form the basis of your editorial assessment.

We would like to point out that our 2016 PLoSOne paper made a theoretical prediction that elevated maternal circulating microRNAs would inhibit placental EMT (epithelial mesenchymal transition). LSA-2018-00252-T actually reports on experiments that test this prediction, and so cannot be characterized as a 'cross-check' of our previous data. It is a test of a prediction, and as such, represents new data.

We think that this data has significance from several perspectives.

1. FASD accounts for between 1 and 5% of school-aged children in the US. It is the single biggest cause of developmental disability. Data on new mechanisms which explain why prenatal ethanol exposure results in growth restriction is clinically important. We think that these studies provide a mechanistic explanation for why prenatal ethanol results in growth deficiencies.

2. We are the first to show an effect of PAE on the placental EMT pathway. Prior to this publication, placental EMT had not been identified as a vulnerable mechanism. Moreover, we assess the influence that placental levels of these miRNAs have on EMT.

3. Most of the identified microRNAs have also been implicated in other diseases of placental insufficiency, so the findings for ethanol effects are likely to be relevant to other causes of placental dysfunction.

4. Most studies on microRNAs deal with the effects of single microRNAs. However, most diseases that involve microRNAs, influence the levels and function of multiple microRNAs. To our knowledge, this is the first study to show that the individual behavior of single microRNAs is different from the collective behavior of a group of microRNAs and that new biology emerges from manipulating an entire group of microRNAs. This itself is a significant finding in the field.

5. A large number of studies have shown the biomarker potential for circulating microRNAs. However, showing that these circulating microRNAs explain and control disease processes represents a new dimension. An analogy might be data showing that elevated plasma cortisol is a marker for stress compared to showing that cortisol results in receptor-mediated alterations in cellular physiology and gene expression. The latter finding moves cortisol from being a marker for stress to being a potential therapeutic target for controlling stress. We feel that the same is true for plasma microRNAs. Evidence for a collective functional role of these microRNAs helps move the field from microRNAs as a marker for disease, to microRNAs as a therapeutic target to protect against disease (in this case, placental insufficiency).

We hope that you will reconsider your decision. If you decide that you need additional reviewer input, we would be happy to provide names of colleagues with whom we have no conflict of interest, who are experts in this field, and whose judgement we respect.

Thank you for your recent correspondence regarding our decision on your work. We have re-assessed your work in light of your arguments and decided to allow you to revise the work for further consideration here. Importantly, the revised manuscript should better highlight the utility of your findings, the figure quality should get improved (reviewer #1) and subsets of HEamiRNAs shared between FASD and other diseases should get tested for affecting placental trophoblast development in vitro (reviewer #2).

[https://lsa.msubmit.net/cgi-](https://lsa.msubmit.net/cgi-bin/main.plex?el=A2Na1Hs3A3CiNe4I6B9ftdNW05GoVCzmyWJfXs7GYLgZ)

[bin/main.plex?el=A2Na1Hs3A3CiNe4I6B9ftdNW05GoVCzmyWJfXs7GYLgZ](https://lsa.msubmit.net/cgi-bin/main.plex?el=A2Na1Hs3A3CiNe4I6B9ftdNW05GoVCzmyWJfXs7GYLgZ)

-- High-resolution figure, supplementary figure and video files uploaded as individual files: See our detailed guidelines for preparing your production-ready images, <http://life-science-alliance.org/authorguide>

B. MANUSCRIPT ORGANIZATION AND FORMATTING:

Full guidelines are available on our Instructions for Authors page, <http://life-science-alliance.org/authorguide>

We are looking forward to receiving your revised manuscript.

Reviewer #1 (Comments to the Authors (Required)):

This is an interesting paper and should be published in some form. There do not appear to be any methodological or design issues and stats are appropriate. The slight lack of enthusiasm comes from the fact that what the authors did was cross-check their human data (PLoSOne 2016) in rodent, non-human primate and in vitro, with a mini review added. The authors come to the same conclusions as from the human study. The utility of the additional findings have not been made clear - leaving the "so what" question for the publication not fully answered. The results are resented in a long-hand way with quite a discursive style and could do with organising better for the reader to follow. The quality of some of the figures in the pdf version I saw was not ideal - too small with sloppy font changes.

I do think that the authors could strengthen the manuscript with a clear address of the clinical relevance of the additional data.

We thank the reviewer for expressing interest in our study. However, we would like to point out that our 2016 PloSOne paper merely made a theoretical prediction that elevated maternal circulating microRNAs would inhibit placental EMT (epithelial mesenchymal transition). In this manuscript we tested this prediction with a set of experiments, and as such it represents new data (lines 45-48 of the modified abstract and lines 415-418 of the modified discussion).

In our revision we have significantly shortened the discussion and portions of the results section to improve readability. For the results section we have shortened and clarified the language in lines 113-119, 127-131, 137-141, 195-200, 224-226, 231, 237, 245-248, 25-268, 306-311, and 328-344. We have also edited both the abstract, introduction and discussion to emphasize the clinical significance and novelty of our study. Specifically:

1. We are the first to show an effect of PAE on the placental EMT pathway. Prior to this publication, placental EMT had not been identified as a vulnerable mechanism. Moreover, we assess the influence that placental levels of these miRNAs have on EMT. We have now highlighted the novelty of this in lines 48-50 of the abstract, lines 97-102 of the introduction, as well as lines 417-421 of the discussion.
2. Most of the identified microRNAs have also been implicated in other diseases of placental insufficiency, so the findings for ethanol effects are likely to be relevant to other causes of placental dysfunction. This is now emphasized in lines 106-109 of the introduction, as well as 451-456 and 461-467 of the discussion
3. Most studies on microRNAs deal with the effects of single microRNAs. However, most diseases that involve microRNAs, influence the levels and function of multiple microRNAs. To our knowledge, this is the first study to show that the individual behavior of single microRNAs is different from the collective behavior of a group of microRNAs and that new biology emerges from manipulating an entire group of microRNAs. This itself is a significant finding in the field. We now include a discussion on this in lines 48-53 of the abstract, 100-102 of the introduction, as well as 419-421 and 477-493 of the discussion.
4. A large number of studies have shown the biomarker potential for circulating microRNAs. However, showing that these circulating microRNAs explain and control disease processes represents a new dimension. An analogy might be data showing that elevated plasma cortisol is a marker for stress compared to showing that cortisol results in receptor-mediated alterations in cellular physiology and gene expression. The latter finding moves cortisol from being a marker

for stress to being a potential therapeutic target for controlling stress. We feel that the same is true for plasma microRNAs. Evidence for a collective functional role of these microRNAs helps move the field from microRNAs as a marker for disease, to microRNAs as a therapeutic target to protect against disease (in this case, placental insufficiency). We have now emphasized this in the concluding paragraph of the discussion section in lines 494-506.

5. FASD accounts for between 1 and 5% of school-aged children in the US. It is the single biggest cause of developmental disability. Data on new mechanisms which explain why prenatal ethanol exposure results in growth restriction is clinically important. We think that these studies provide a mechanistic explanation for why prenatal ethanol results in growth deficiencies. This is now emphasized in lines 59-64 of the introduction.

We have also corrected sizing problems and standardized font usage across all figures and supplemental figures. These figures should now also be uploaded as individual high-resolution tiff images.

Reviewer #2 (Comments to the Authors (Required)):

This paper focused on miRNAs, the expressions of which are significantly elevated in the circulation of pregnant mothers who have infants affected by prenatal alcohol exposure in their Ukrainian cohort. The overexpression of miRNAs disrupts placental development, suggesting that these miRNAs may contribute to the pathology of Fetal Alcohol Spectrum Disorders (FASD). The study used not only rodent and primate models but also in vitro models of human placental trophoblast development.

Although the direct targets of these miRNAs are still elusive, I think this manuscript indicates nicely that the set of miRNAs defined in human patients is potentially involved in pathogenesis in FASD. Their data will be foundation of the mechanisms of FASD placenta pathology and may lead to novel intervention that ameliorates FASD symptoms.

Major comment:

The manuscript showed overlapping miRNAs between FASD and other diseases such as preeclampsia, spontaneous abortion, fetal growth restriction etc. Since impairment of trophoblast EMT is foundation of these diseases and the 11 HEamiRNAs collectively prevent this process, I wonder if subsets of HEamiRNAs shared between FASD and other diseases affect placental trophoblast development in vitro. Can authors add comments on it?

We thank the reviewer for expressing interest in our study. We have now looked at the effect of subsets of HEamiRNAs shared between FASD and other gestation pathologies on cytotrophoblast EMT. These data are now presented in Supplementary figure 6 as well as lines 211-222 of the results:

“We next sought to determine if more restricted subsets of HEamiRNAs could recapitulate the effects of HEamiRNAs collectively on EMT. Thus, we overexpressed hsa-miR-222-5p and hsa-miR-519a-3p, which are implicated in preeclampsia and fetal growth restriction, as well as hsa-miR-885-3p, hsa-miR-518f-3p, hsa-miR-204-5p, which are implicated in preeclampsia, fetal growth restriction, and spontaneous abortion or preterm labor (Supplementary Figure 6A). In contrast to the collective action for all HEamiRNAs, exposure to each of these pools resulted in significant decreases in CDH1 expression ($F(2,12)=20.12$, $p=0.0001$). The pool including hsa-miR-885-3p, hsa-miR-518f-3p, hsa-miR-204-5p also significantly increased Snai1 ($F(2,12)=4.604$, $p=0.0328$; Dunnett’s post-hoc $p=0.0497$, Supplementary Figure 6B-E). These data suggest that HEamiRNAs include sub-groups of miRNAs that have the potential

to partly mitigate the effects of elevating the entire pool. However, the potential protective effects of these sub-groups are masked by the collective function of the entire group of HEamiRNAs.”

We have also added comments on these results in lines 461-467 of the discussion:

“We also observed that overexpression of more restricted subsets of HEamiRNAs associated with preeclampsia, fetal growth restriction, and spontaneous abortion or preterm labor also partly promoted EMT transcript signatures, contrasting with the collective inhibitory action of HEamiRNAs as a whole. Thus, elevation of some subsets of HEamiRNAs may constitute a compensatory mechanism aimed at minimizing placental pathologies, though their potential protective effects are masked by the collective elevation of HEamiRNAs.”

And lines 104-109 of the introduction:

“Collectively, these data suggest that elevated HEamiRNAs may represent an emergent maternal stress response that triggers fetal growth restriction, though sub-groups of HEamiRNAs may compete to protect against the loss of EMT. Moreover, most members of the group of HEamiRNAs, have also been implicated in other placental insufficiency and growth restriction syndromes, giving rise to the possibility that growth restriction syndromes may share common etiological mediators”

February 19, 2019

RE: Life Science Alliance Manuscript #LSA-2018-00252-TR-A

Dr. Rajesh C Miranda
Texas A&M University Health Science Center
Neuroscience and Experimental Therapeutics
8447 Riverside Pkwy, MREB Suite 1005
Bryan, TX 77807

Dear Dr. Miranda,

Thank you for submitting your revised manuscript entitled "Maternal Circulating MiRNAs That Predict Infant FASD Outcomes Influence Placental Maturation". We appreciate the introduced changes and we would be happy to publish your paper in Life Science Alliance pending final revisions necessary to meet our formatting guidelines:

- please add scale bars in figure 2
- please list 10 authors et al in your reference list
- please add a description for panel G in the figure legend for figure 10
- please add labels for the excel tables so that the typesetters know which table they are looking at

A. FINAL FILES:

-- Summary blurb (enter in submission system): A short text summarizing in a single sentence the study (max. 200 characters including spaces). This text is used in conjunction with the titles of papers, hence should be informative and complementary to the title. It should describe the context and significance of the findings for a general readership; it should be written in the present tense

and refer to the work in the third person. Author names should not be mentioned.

B. MANUSCRIPT ORGANIZATION AND FORMATTING:

Sincerely,

February 21, 2019

RE: Life Science Alliance Manuscript #LSA-2018-00252-TRR

Dr. Rajesh C Miranda
Texas A&M University Health Science Center
Neuroscience and Experimental Therapeutics
8447 Riverside Pkwy, MREB Suite 1005
Bryan, TX 77807

Dear Dr. Miranda,

Thank you for submitting your Research Article entitled "Maternal Circulating MiRNAs That Predict Infant FASD Outcomes Influence Placental Maturation". It is a pleasure to let you know that your manuscript is now accepted for publication in Life Science Alliance. Congratulations on this interesting work.

DISTRIBUTION OF MATERIALS:

Again, congratulations on a very nice paper. I hope you found the review process to be constructive and are pleased with how the manuscript was handled editorially. We look forward to future exciting submissions from your lab.